# Continuous-Time Analysis of Heavy Ball Momentum in Min-Max Games

**Yi Feng** [1]   **Kaito Fujii** [2]   **Stratis Skoulakis** [3]   **Xiao Wang** [1] [4]   **Volkan Cevher** [5]

## Abstract

Since Polyak's pioneering work, heavy ball (HB) momentum has been widely studied in minimization. However, its role in min-max games remains largely unexplored. As a key component of practical min-max algorithms like Adam, this gap limits their effectiveness. In this paper, we present a continuous-time analysis for HB with simultaneous and alternating update schemes in min-max games. Locally, we prove *smaller* momentum enhances algorithmic stability by enabling local convergence across a wider range of step sizes, with alternating updates generally converging faster. Globally, we study the implicit regularization of HB, and find *smaller* momentum guides algorithms trajectories towards shallower slope regions of the loss landscapes, with alternating updates amplifying this effect. Surprisingly, all these phenomena differ from those observed in minimization, where *larger* momentum yields similar effects. Our results reveal fundamental differences between HB in min-max games and minimization, and numerical experiments further validate our theoretical results.

## 1. Introduction

The Heavy Ball momentum (a.k.a. Polyak's momentum) (Polyak, 1964) has played an important role in modern optimization for machine learning (Sutskever et al., 2013). In the context of minimization, the mechanism of heavy ball momentum is typically explained by modeling it as a continuous-time differential equation that describes the physical process of a mass moving through a frictional medium (Qian, 1999). Due to its intuitive appeal and the

valuable insights it provides into the behavior of discrete-time algorithms, the approach of using continuous-time equations to analyze momentum-based methods in minimization has seen significant advancements in recent years (Su et al., 2016; Kovachki & Stuart, 2021; Shi et al., 2022).

Min-max games model optimization scenarios where two players interact such that one's gain is the other's loss, with the objective of finding a local Nash equilibrium. They are widely applied in tasks like GANs (Goodfellow et al., 2014) and adversarial training (Madry, 2017). Due to their extreme importance, many specialized algorithms like Extra-gradient (Korpelevich, 1976) and optimistic methods (Daskalakis et al., 2017) have been proposed to solve this type of problems. However, heavy ball momentum, as a crucial component of practical min-max algorithms such as Adam, is extensively employed in solving min-max games (Wang et al., 2019; Do et al., 2022). Our understanding of heavy ball momentum in min-max games remains relatively limited compared to minimization.

For example, Gidel et al. (2019) observed that in min-max games, negative momentum can enhance training outcomes, which is surprising given that positive momentum is typically employed in minimization. Moreover, in min-max games, players can update their strategies simultaneously, as in repeated Rock-Paper-Scissors, or alternately, as in GANs training, where the discriminator is updated firstly, and then the generator updates based on the feedback it received from the discriminator. It turns out that this subtle difference on the update rules can greatly affect the results of learning process, and Gidel et al. (2019) proved in the special cases of bilinear games, only combining negative momentum with the alternating updates can lead to convergence. These discoveries motivate the following questions:

*What is the impact of momentum and update rules on the learning dynamics in min-max games?*

In this paper, we investigate min-max learning dynamics through the lens of continuous-time analysis, focusing on both *local* and *global* aspects. Locally, we quantitatively analyze how momentum and update rules influence the convergence rates of continuous-time models near a local Nash equilibrium—a crucial first step in understanding algorithmic behavior (Ochs, 2018). Globally, given the limited understanding of heavy ball momentum's properties even

---

The first four authors are listed in alphabetical order. [1]Shanghai University of Finance and Economics, Shanghai, China [2]National Institute of Informatics, Tokyo, Japan [3]Aarhus University, Aarhus, Denmark [4]Key Laboratory of Interdisciplinary Research of Computation and Economics, China [5]EPFL, Lausanne, Switzerland. Correspondence to: Xiao Wang <wangxiao@sufe.edu.cn>.

*Proceedings of the 42$^{nd}$ International Conference on Machine Learning*, Vancouver, Canada. PMLR 267, 2025. Copyright 2025 by the author(s).

in minimization (Lessard et al., 2016; Wang et al., 2022), we take a qualitative approach to explore global dynamics. In minimization, such global dynamics can be interpreted through implicit gradient regularization (Barrett & Dherin, 2021), which describes how optimization trajectories interact with loss landscape geometry. We extend this perspective to heavy ball momentum in min-max games.

## 1.1. Summary of Contributions

In this paper, we investigate local convergence and implicit gradient regularization of heavy ball momentum in min-max games through their continuous-time models. Our main contributions can be summarized as:

- **Local Convergence**: Our results highlight two insights. Firstly, there exists a trade-off between positive and negative momentum: positive momentum achieves an optimal local convergence rate under small step sizes (Theorem 4.8), whereas negative momentum enables convergence across a broader range of step sizes (Corollary 4.7). This contrasts with minimization, where large positive momentum can facilitate local convergence over a broader range of step sizes (O'Donoghue & Candes, 2015). Secondly, we establish conditions under which alternating updates outperform simultaneous ones (Theorem 4.9). These conditions capture important min-max game scenarios where players' optimization dynamics have strong interaction, including bilinear games in (Gidel et al., 2019).

- **Implicit Gradient Regularization**: Our analysis of continuous-time models reveals that *smaller* momentum and alternating updates steer algorithm trajectories toward regions with shallower slopes in the min-max loss landscape, characterized by smaller gradient norms. Interestingly, this contrasts with minimization problems, where *larger* momentum facilitates convergence to shallow slope regions (Ghosh et al., 2023). To validate these findings, we conduct numerical experiments exploring various dynamical behaviors, including convergence to equilibrium, limit cycles, and GAN training dynamics. Furthermore, since sharp gradient regions are a major source of instability in min-max training (Thanh-Tung et al., 2019; Wu et al., 2021), our results provide an explanation for the observed benefits of negative momentum in improving stability, as noted in (Gidel et al., 2019).

## 1.2. Related Works

**Momentum in Games Dynamics.** The role of momentum in min-max games has received considerable attention. (Gidel et al., 2019) empirically observed that negative momentum enhances min-max training and established its theoretical benefits in bilinear games. (Zhang & Wang, 2021) investigated the local convergence rate of *simultaneous* HB in

strongly-convex-strongly-concave (SCSC) min-max games, highlighting its suboptimality, the result later extended to global convergence under the same condition by (Zhang et al., 2021). (Azizian et al., 2020) demonstrated that HB accelerates local convergence when the underlying games closely resemble to minimization problems. (Fang et al., 2025) investigated the no-regret properties of heavy ball momentum. (Lotidis et al., 2024) examined Nesterov's momentum in general games with strict equilibrium. Our work differs from the studies above. We do not assume SCSC conditions and instead focus on more representative min-max games, where strong player interactions fundamentally distinguish them from minimization. Furthermore, our goal is *not* to prove that HB achieves a better convergence rate than other algorithms like Extra-gradient, which has been shown to be impossible by (Azizian et al., 2020). Instead, we aim to explore how the parameters of momentum algorithms influence the dynamics of min-max games.

**Alternating Updates in Game Dynamics.** Recent studies have demonstrated the benefits of alternating updates in various game-theoretic settings. (Bailey et al., 2020; Wibisono et al., 2022; Hait et al., 2025) showed that alternating updates lead to a slower regret growth rate from an online learning perspective. (Feng et al., 2024) studied the complex dynamical behaviors of alternating updates from the prediction accuracy perspective. (Rosca et al., 2021) investigated different discretization drifts arising from simultaneous and alternating GDA, i.e., algorithm without momentum. (Yang et al., 2022) showed that alternating GDA achieves a faster convergence rate under PL condition. (Zhang et al., 2022; Lee et al., 2024) proved that alternating GDA converges more rapidly than simultaneous GDA in SCSC settings. To the best of our knowledge, the convergence separation result between simultaneous and alternating HB in bilinear games (Gidel et al., 2019) is the only known theoretical result on alternating HB.

**Differential Equations for Optimization.** Differential equations are powerful tools for studying optimization algorithms. In the following, we present a very brief summary, and future related works can be found in the references therein. For minimization, (Su et al., 2016; Wibisono et al., 2016) purposed ODEs to investigate momentum in convex minimization. The use of SDEs for the stochastic setting was developed by (Li et al., 2017) and future developed by (Compagnoni et al., 2024a) recently. For min-max games, (Lu, 2022) introduced methods for deriving high resolution ODEs to study algorithms' convergence behaviors. (Hsieh et al., 2021) purposed mean dynamics for Robbins–Monro algorithms to study algorithms' long-time behaviors. For the stochastic setting, recent work of (Compagnoni et al., 2024b) derived SDEs to model the behaviors of several min-max algorithms under the weak approximation framework.

### 1.3. Organization

In Section 2, we outline the background for this paper. Section 3 introduces our continuous-time models for momentum algorithms. In Section 4, we present the local analysis results, followed by discussions on implicit gradient regularization in Section 5. Numerical experiments are integrated into the main text, while additional experiments and proofs are provided in the appendix due to space constraints.

## 2. Preliminaries

**Min-Max Games.** A min-max game with smooth payoff function $f(x, y)$ can be formulated as

$$\min_{x \in \mathbb{R}^n} \max_{y \in \mathbb{R}^m} f(x, y) \qquad \text{(Min-Max Games)}$$

If a pair of strategies $(x^*, y^*)$ satisfies $\forall x \in \mathcal{U},\ f(x, y^*) \geq f(x^*, y^*)$ and $\forall y \in \mathcal{V},\ f(x^*, y^*) \geq f(x^*, y)$ for some $x^*$'s neighborhood $\mathcal{U} \subseteq \mathbb{R}^n$ and $y^*$'s neighborhood $\mathcal{V} \subseteq \mathbb{R}^m$, then $(x^*, y^*)$ is called a *local Nash equilibrium*, which is a standard solution concept in min-max games (Ratliff et al., 2016).

**Heavy Ball Momentum.** In this paper, we both consider the simultaneous heavy ball momentum method:

$$x_{n+1} = x_n - h\nabla_x f(x_n, y_n) + \beta(x_n - x_{n-1})$$
$$y_{n+1} = y_n + h\nabla_y f(x_n, y_n) + \beta(y_n - y_{n-1}) \quad \text{(Sim-HB)}$$

and the alternating heavy ball momentum method:

$$x_{n+1} = x_n - h\nabla_x f(x_n, y_n) + \beta(x_n - x_{n-1})$$
$$y_{n+1} = y_n + h\nabla_y f(x_{n+1}, y_n) + \beta(y_n - y_{n-1})$$
$$\text{(Alt-HB)}$$

Here $\beta \in (-1, 1)$ is the momentum parameter and $h > 0$ is the step size. We refer to $\beta < 0$ as negative momentum. We use the word "smaller momentum" to mean that the value of $\beta$ is smaller, not its absolute value.

**Local Behaviors of Dynamical System.** For a system of differential equations $\dot{x}(t) = g(x)$ where $g : \mathbb{R}^n \to \mathbb{R}^n$ is a differentiable function, let $\tilde{x} \in \mathbb{R}^n$ satisfy $g(\tilde{x}) = 0$. Then the local behavior of the system near $\tilde{x}$ is determined by the eigenvalues of Jacobian $\mathcal{J}_g(\tilde{x}) = \left(\frac{\partial g_i}{\partial x_j}(\tilde{x})\right)_{i,j}$. In particular, we have the following standard result:

**Proposition 2.1** (Muehlebach & Jordan (2021))**.** *If $\alpha = \max_{\lambda \in \mathrm{Sp}(\mathcal{J}_g)} \Re(\lambda) < 0$, then there exist constants $\delta > 0$ and $C > 0$ such that for all initial conditions satisfying $\|x(0) - \tilde{x}\| \leq \delta$, we have $\|x(t) - \tilde{x}\| \leq Ce^{t\alpha}, \forall t > 0$.*

**Notation.** We denote the set of real numbers by $\mathbb{R}$ and the set of complex numbers by $\mathbb{C}$. For any matrix $\mathcal{M} \in \mathbb{R}^{d \times d}$

or $\mathcal{M} \in \mathbb{C}^{d \times d}$, we use $\mathrm{Sp}(\mathcal{M})$ to represent the set of its eigenvalues in $\mathbb{C}$. Given $\lambda \in \mathrm{Sp}(\mathcal{M})$, we use $\Re(\lambda)$ and $\Im(\lambda)$ to denote the real and imaginary parts of $\lambda$, respectively. The notation $\mathcal{M} \preccurlyeq \mathbf{0}$ or $\mathcal{M} \succcurlyeq \mathbf{0}$ indicates that $\mathcal{M}$ is a negative or positive semi-definite matrix, respectively. We use $\mathrm{EigVec}(\mathcal{M})$ to denote the eigenspace of $\mathcal{M}$, and $\mathrm{Ker}(\mathcal{M})$ to represent the kernel space of $\mathcal{M}$, i.e., $\mathrm{Ker}(\mathcal{M}) = \{z \in \mathbb{C}^d \mid \mathcal{M}z = \mathbf{0}\}$.

## 3. Continuous-Time Models

Continuous-time models have proven to be effective in analyzing momentum-based algorithms in minimization problems. We extend this approach to heavy ball momentum algorithms in min-max games. We firstly define the modified loss function $\mathcal{F}(x, y)$ as follows:

$$\mathcal{F}(x, y) = \left(\frac{1}{1 - \beta}\right) f(x, y)$$
$$+ \frac{h(1 + \beta)}{4(1 - \beta)^3} \left(\|\nabla_x f(x, y)\|^2 - \|\nabla_y f(x, y)\|^2\right).$$

For Sim-HB, the continuous-time models are given by:

$$\dot{x}(t) = -\nabla_x \mathcal{F}(x, y), \quad \dot{y}(t) = \nabla_y \mathcal{F}(x, y).$$
$$\text{(Continuous Sim-HB)}$$

For Alt-HB, the continuous-time models are:

$$\dot{y}(t) = \nabla_y \left(\mathcal{F}(x, y) - \frac{h}{2(1 - \beta)^2}\|\nabla_x f(x, y)\|^2\right),$$
$$\dot{x}(t) = -\nabla_x \mathcal{F}(x, y). \qquad \text{(Continuous Alt-HB)}$$

It may seem surprising that our equations are first-order rather than the commonly used second-order equations for modeling momentum in minimization (Su et al., 2016). However, we chose not to adopt their approach for two reasons. First, their method couples the step sizes and momentum parameter, which hinders the independent analysis of the impact of each term. Second, their approach requires $\beta$ to be positive to align with physical intuition, whereas our goal is to also explore algorithms with negative momentum.

The derivation of our equations is inspired by (Ghosh et al., 2023), which employed continuous-time models to study the implicit regularization of the heavy ball method in minimization. However, our equations differ significantly due to the presence of two interacting players and the distinct update sequences inherent to min-max games.

We provide analysis of the approximation error between continuous-time equations and original algorithms in the following Theorem. The proof is deferred to Appendix B.

**Theorem 3.1.** *Let $f(x, y)$ be a smooth function with bounded derivatives up to the fourth order. Then for*

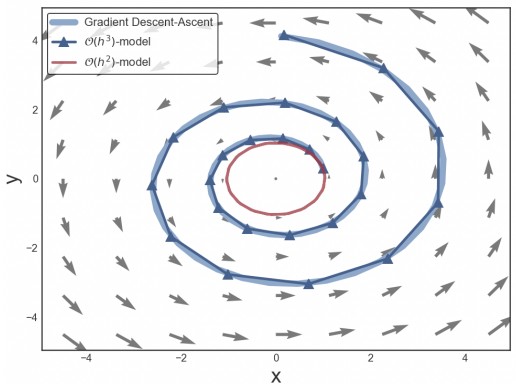

*Figure 1.* Comparison of $\mathcal{O}(h^3)$ and $\mathcal{O}(h^2)$-local error models with payoff function $f(x,y) = xy$.

*any given finite time horizon, the solution trajectories $(x(t), y(t))$ of Continuous Sim-HB (resp. Continuous Alt-HB) is locally $\mathcal{O}(h^3)$-close to the trajectories of Sim-HB (resp. Alt-HB) after $4 \log h / \log|\beta|$ steps.*

To highlight the advantages of our $\mathcal{O}(h^3)$ local error models, we compare them with existing continuous-time approximations of the gradient descent-ascent (GDA) method ($\beta = 0$ case). The $\mathcal{O}(h^2)$ model proposed by (Bailey et al., 2020) introduces larger local errors, failing to accurately capture GDA dynamics. For example, while simultaneous GDA is known to diverge in bilinear games, their model incorrectly predicts cyclic behavior. In contrast, as illustrated in Figure 1, the $\mathcal{O}(h^3)$ models address these shortcomings. Future experiments ar presented in Appendix A.

Our continuous-time models also have strong consistency with prior works. First, (Rosca et al., 2021) purposed continuous-time equations to model the dynamics of GDA algorithm. Notably, when $\beta = 0$, our equations reduce to their formulation. Second, (Gidel et al., 2019) proved that in bilinear games, Alt-HB with a negative $\beta$ converges to equilibrium at an exponential rate, whereas Sim-HB diverges regardless the choice of $h$ and $\beta$. In Proposition 3.2, we show that our continuous-time equations yield the same conclusion. The proof is provided in Appendix C.

**Proposition 3.2** (Consistency with Gidel et al. (2019)). *In matrix games $f(x,y) = x^\top A y$, Continuous Sim-HB diverges at an exponential rate, regardless of the values of $h$ and $\beta$. In contrast, Continuous Alt-HB converges exponentially when $\beta$ is negative and $h$ is sufficiently small.*

# 4. Analysis of Local Dynamics

In this section, we analyze the local behavior of the continuous-time equations introduced in Section 3 near equilibrium. Additional numerical experiments validating our theoretical findings and demonstrating the accuracy of

continuous-time models in predicting discrete-time dynamics are provided in Appendix I.

## 4.1. Jacobian of Continuous-time Models

This section examines the Jacobian of the continuous-time models. As demonstrated in Proposition 2.1, the Jacobian matrices play a crucial role in analyzing the local behavior of these equations. We start by presenting the Jacobian for the Gradient Descent-Ascent flow, given by:

$$\dot{x} = -\nabla_x f(x,y), \ \dot{y} = \nabla_y f(x,y) \quad \text{(Gradient Flow)}$$

**Lemma 4.1.** *The Jacobian of Gradient Flow at a local equilibrium point $(x^*, y^*)$ is given by:*

$$\mathcal{J} = \begin{bmatrix} -\nabla_x^2 f(x^*, y^*) & -\nabla_{xy} f(x^*, y^*) \\ \nabla_{yx} f(x^*, y^*) & \nabla_y^2 f(x^*, y^*) \end{bmatrix} \quad \text{(Jacobian)}$$

*By the definition of local Nash equilibrium, we also have*

$$\nabla_x^2 f(x^*, y^*) \succcurlyeq \mathbf{0}, \ \nabla_y^2 f(x^*, y^*) \preccurlyeq \mathbf{0}.$$

We now present the Jacobian of Continuous Sim-HB and Continuous Alt-HB at equilibrium. The detailed calculations are deferred to Appendix D.

**Proposition 4.2** (Jacobian for simultaneous updates). *Let $\mathcal{J}_S$ denote the Jacobian of Continuous Sim-HB at the local equilibrium $(x^*, y^*)$. Then, $\mathcal{J}_S$ can be expressed as a quadratic polynomial in terms of (Jacobian):*

$$\mathcal{J}_S = \left( \frac{1}{1-\beta} \mathcal{I} - \frac{h(1+\beta)}{2(1-\beta)^3} \mathcal{J} \right) \cdot \mathcal{J}$$

*where $\mathcal{I}$ denotes the identity matrix.*

**Proposition 4.3** (Jacobian for alternating updates). *Let $\mathcal{J}_A$ denote the Jacobian of Continuous Alt-HB at the local equilibrium $(x^*, y^*)$. Then, $\mathcal{J}_A$ can be expressed as*

$$\mathcal{J}_A = \mathcal{J}_S - \frac{h}{(1-\beta)^2} \begin{bmatrix} \mathbf{0} & \mathbf{0} \\ \nabla_{yx} f \nabla_x^2 f & \nabla_{yx} f \nabla_{xy} f \end{bmatrix}_{(x^*, y^*)}$$

In Sections 4.2 and 4.3, we use these Jacobian matrices to study the local convergence behavior of the continuous-time models. To facilitate this analysis, we first revisit the decomposition of (Jacobian) introduced in (Letcher et al., 2019):

$$\mathcal{J} = \mathcal{S} + \mathcal{A} \quad \text{(Decomposition)}$$

where

$$\mathcal{S} = \begin{bmatrix} -\nabla_x^2 f(x^*, y^*) & \mathbf{0} \\ \mathbf{0}^\top & \nabla_y^2 f(x^*, y^*) \end{bmatrix} \quad \text{(Potential)}$$

is called the *potential part* of the game dynamics, and

$$
\mathcal{A} = \begin{bmatrix} \mathbf{0} & -\nabla_{xy} f(x^*, y^*) \\ \nabla_{yx} f(x^*, y^*) & \mathbf{0}^\top \end{bmatrix} \quad \text{(Hamiltonian)}
$$

is called the *Hamiltonian part*. Recall that the local structure of a game near an equilibrium is essentially captured by the quadratic form:

$$
\frac{1}{2} \tilde{x}^\top \nabla_x^2 f(x^*, y^*) \tilde{x} + \frac{1}{2} \tilde{y}^\top \nabla_y^2 f(x^*, y^*) \tilde{y}
$$
$$
+ \tilde{x}^\top \nabla_{xy} f(x^*, y^*) \tilde{y}, \quad (1)
$$

where $\tilde{x} = x - x^*$, $\tilde{y} = y - y^*$. From (1), (Potential) represents the non-interactive aspects of player behavior, while (Hamiltonian) captures player interactions. In the context of min-max games, the most interesting scenarios arise when there is strong interaction between the two players, i.e., $\mathcal{A} \gg \mathcal{S}$, which is the focus of this paper. This contrasts with cases where players independently optimize their objective functions, such as in games with payoff function $f(x, y) = f_1(x) + f_2(y)$, where $\mathcal{A} = \mathbf{0}$. We formulate this in the following assumption:

**Assumption 4.4.** *In the following, we assume $\mathcal{A} \gg \mathcal{S}$ in (Decomposition). Since $\mathcal{A}$ is an antisymmetric matrix and has purely imaginary eigenvalues, $\mathcal{A} \gg \mathcal{S}$ implies $|\Im(\lambda)| > |\Re(\lambda)|, \forall \lambda \in \mathrm{Sp}(\mathcal{J})$.*

### 4.2. Analysis of Simultaneous Updates

In this section, we study the local behavior of Continuous Sim-HB. Unlike minimization, where local convergence of algorithms is usually guaranteed, the scenario for min-max games is more complex. As shown in Proposition 3.2, heavy ball momentum may fail in bilinear games. This observation leads to our first question:

*Does the failure of Continuous Sim-HB in bilinear games indicate a general limitation of this method, or is it simply because bilinear games are not representative?*

Furthermore, Gidel et al. (2019) observe that **negative** momentum can significantly enhance training performance in practice. This is in sharp contrast to minimization, where **positive** momentum is used to accelerate convergence. This observation motivates our second question:

*What are the respective benefits of using Continuous Sim-HB with negative versus positive momentum for local dynamics?*

Our answers to above questions can be summarized as:

- The failure of Continuous Sim-HB in bilinear games is not representative. In Theorem 4.6, we prove that simultaneous heavy ball momentum generically achieves local convergence under appropriate parameter choices.

- Negative momentum and positive momentum benefit game dynamics in distinct ways. Negative momentum enhances algorithm stability by enabling convergence even with large step sizes (Corollary 4.7). Conversely, if a sufficiently small step size $h$ is employed, positive momentum can achieve an optimal convergence rate (Theorem 4.8).

We first recall the following assumption from (Wang & Chizat, 2024), which is used to guarantee the local convergence of (Gradient Flow) in their work:

**Assumption 4.5.** *In* (Decomposition)*, we assume*

$$
\mathrm{EigVec}(\mathcal{A}) \cap \mathrm{Ker}(\mathcal{S}) = \{\mathbf{0}\}.
$$

Assumption 4.5 holds **generically** in the following sense: for any fixed $\mathcal{S} \neq \mathbf{0}$, if $\mathcal{A}$ is sampled independently from an absolutely continuous distribution, then Assumption 4.5 holds with probability one. Compared to the strongly-convex-strongly-concave setting, which requires *all* eigenvalues of block matrices of $\mathcal{S}$ to be strictly negative, Assumption 4.5 permits *many* eigenvalues of these matrices to be zero. This significantly broadens the applicability beyond the strongly-convex-strongly-concave framework.

**Theorem 4.6.** *Let $\mathcal{J}$ be the* (Jacobian) *defined in Lemma 4.1. If the momentum parameter $\beta \in (-1, 1)$ and the step size $h > 0$ satisfy the inequality*

$$
h < \min_{\lambda \in \mathrm{Sp}(\mathcal{J})} \frac{2(1 - \beta)^2}{(1 + \beta)} \frac{|\Re(\lambda)|}{(\Im(\lambda)^2 - \Re(\lambda)^2)} \quad (2)
$$

*and if $f(x, y)$ satisfies Assumption 4.4 and 4.5 at the local equilibrium $(x^*, y^*)$, then Continuous Sim-HB achieves local convergence.*

Since Assumption 4.4 captures the critical scenarios where player interactions dominate game dynamics, and Assumption 4.5 holds generically, Theorem 4.6 implies that Continuous Sim-HB achieves local convergence under fairly mild conditions. Furthermore, the bilinear games studied in Proposition 3.2 do not satisfy Assumption 4.5 as $\mathcal{S}$ is always a zero matrix in that case.

The proof of Theorem 4.6 is based on the analysis of the Jacobian of Continuous Sim-HB. By Proposition 4.2, this Jacobian can be expressed as a quadratic polynomial in terms of $\mathcal{J}$, establishing a connection between $\mathrm{Sp}(\mathcal{J}_S)$ and $\mathrm{Sp}(\mathcal{J})$, which simplifies the analysis. The detailed proof is deferred to the Appendix E.

**Corollary 4.7.** *Since the function $2(1 - \beta)^2/(1 + \beta)$ in Theorem 4.6 is a decreasing function of $\beta \in (-1, 1)$, a smaller value of $\beta \in (-1, 1)$ enables local convergence of Continuous Sim-HB across a broader range of step sizes.*

Corollary 4.7 highlights an interesting contrast between the role of momentum in min-max games and in minimization.

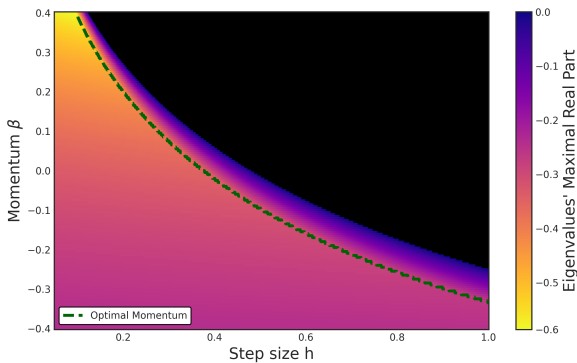

*Figure 2.* Distribution on the eigenvalues' maximal real part of $\mathcal{J}_S$, which governs the local behaviors according to Proposition 2.1. The **black** region indicates divergence for the corresponding parameters. Smaller momentum expands the range of step sizes for convergence, supporting Corollary 4.7. For small step sizes, the optimal momentum is positive, consistent with Theorem 4.8.

In minimization, it is well known that *larger* momentum can enable local convergence with a broader range of step sizes. We also provide proof and experiments on this point in Appendix F.

However, Corollary 4.7 does not imply that the smallest possible momentum should always be used. Theorem 4.8 shows that excessively negative momentum can slow convergence. Additionally, for sufficiently small step sizes, the optimal momentum is positive.

**Theorem 4.8.** *If $f(x, y)$ satisfies Assumption 4.4 and 4.5, and if $h \leq \min_{\lambda \in \mathrm{Sp}(\mathcal{J})} \frac{|\Re(\lambda)|}{2(\Im(\lambda)^2 - \Re(\lambda)^2)}$, then the momentum parameter $\tilde{\beta}(h)$ that achieves optimal local convergence rate for* Continuous Sim-HB *is a positive number.*

Theorem 4.8 is proved by analyzing the derivative of the eigenvalue of $\mathcal{J}_S$ with respect to the momentum parameter $\beta$ and examining the location of the roots of this derivative. The detailed proof is provided in Appendix G.

In Figure 2, we presents numerical experiments validating the theoretical results in this section. The findings show that smaller momentum allows the algorithm to achieve local convergence over a broader range of step sizes. Additionally, for sufficiently small step sizes, positive momentum enables the algorithm to achieve the optimal convergence rate. These results align with the theoretical predictions in this section.

### 4.3. Analysis of Alternating Updates

In this section, we investigate the convergence rate of Continuous Alt-HB near a local Nash equilibrium $(x^*, y^*)$. Specifically, drawing on the observation from Proposition 3.2, which highlights the separation between Continuous Sim-HB and Continuous Alt-HB in bilinear games, we

ask the following questions:

*Given the same step size $h$ and momentum $\beta$, which types of games enable* Continuous Alt-HB *to achieve a better convergence rate compared to* Continuous Sim-HB*? Moreover, how can we quantitatively characterize the difference in their convergence rates?*

To answer above questions, we first present a more refined version of Assumption 4.4. Especially, for the Jacobian decomposition $\mathcal{J} = \mathcal{S} + \mathcal{A}$ from (Decomposition), we introduce a small nonnegative number $\alpha \geq 0$ to scale $\mathcal{S}$, leading to the refined version of (Decomposition):

$$
\mathcal{J} = \alpha \begin{bmatrix} -\mathcal{Q} & 0 \\ 0^\top & \mathcal{R} \end{bmatrix} + \begin{bmatrix} \mathbf{0} & -\nabla_{xy}f \\ \nabla_{yx}f & \mathbf{0} \end{bmatrix}_{(x^*, y^*)} \quad (3)
$$

where $\alpha\mathcal{Q} = \nabla_x^2 f(x^*, y^*)$, and $\alpha\mathcal{R} = \nabla_y^2 f(x^*, y^*)$. This decomposition is also utilized to establish the convergence rate of (Gradient Flow) in (Wang & Chizat, 2024).

In Theorem 4.9, we characterize the largest real part of the eigenvalues of $\mathcal{J}_A$, the Jacobian of Continuous Alt-HB. According to Proposition 2.1, a smaller real part implies a faster convergence rate. Additionally, for technical reasons, we assume $m = n$ for (Min-Max Games).

**Theorem 4.9.** *Suppose $\nabla_{xy}f(x^*, y^*)$ is full-rank and has distinct singular values. Let $\lambda'_A$ be the eigenvalue of $\mathcal{J}_A$ with the largest real part. Then there exist an eigenvalue $\lambda_S$ of $\mathcal{J}_S$ and a singular value $\sigma$ of $\nabla_{xy}f(x^*, y^*)$ such that*

$$
\Re(\lambda'_A) = \Re(\lambda_S) - \frac{h}{(1-\beta)^2}|\sigma|^2 + \mathcal{O}(h^2 + h\alpha). \quad (4)
$$

Theorem 4.9 demonstrates that when $h$ and $\alpha$ are sufficiently small, such that the term $\mathcal{O}(h^2 + h\alpha)$ becomes negligible, Continuous Alt-HB exhibits an exponentially accelerated local convergence rate compared to Continuous Sim-HB. The algebraic condition for $\nabla_{xy}f(x^*, y^*)$ is mild, as it holds with probability one for randomly generated matrices from absolutely continuous distribution. The numerical experiments for Theorem 4.9 are provided in Appendix I.3.

To establish Theorem 4.9, we regard $\mathcal{J}_A$ as perturbation of $\mathcal{J}_S$ with respect to parameters $\alpha$ and $h$. The theorem can be proved using tools from matrix perturbation theory. The detailed proof is provided in Appendix H. Additionally, computing the coefficients of $\mathcal{O}(h\alpha)$ in (4) requires calculating perturbed normalized eigenvectors, which is widely recognized as a challenging problem (Bamieh, 2020), and we choose to leave this topic for further work.

**Remark 4.10.** *The requirement that $h$ and $\alpha$ in Theorem 4.9 be sufficiently small is crucial for establishing the superiority of* Continuous Alt-HB*. Appendix I.4 presents an example illustrating that this superiority may not hold if the condition is not satisfied.*

## 5. Implicit Gradient Regularization

Recent studies in minimization have shown that continuous-time models can reveal how algorithms like gradient descent implicitly regularize optimization trajectories, guiding them toward shallower slope regions of the loss landscape, i.e., areas with smaller $\|\nabla f(x)\|^2$ when $f(x)$ is the objective function. This phenomenon, known as *implicit gradient regularization* (Barrett & Dherin, 2021), is crucial for understanding the effectiveness of optimization algorithms in modern deep learning, as flatter solutions are linked to better generalization (Foret et al., 2021). Additionally, (Ghosh et al., 2023) investigated the implicit gradient regularization of heavy ball momentum algorithm in minimization and demonstrated that a ***larger*** momentum parameter $\beta$ can amplify this effect. Motivated by these findings, a natural question arises:

*How does momentum influence the interaction between algorithm trajectories and the shallow slope regions of the loss landscapes in min-max games?*

This section aims to argue the following thesis, highlighting that the phenomenon in min-max games is opposite to that in minimization (Ghosh et al., 2023):

**Thesis.** In min-max games where players' interactions dominate the overall dynamics as described in Assumption 4.4, ***smaller*** momentum guides algorithms' trajectories towards shallower slope regions of loss landscapes, i.e., regions with lower values of $\|\nabla_x f\|^2 + \|\nabla_y f\|^2$. Moreover, alternating updates amplify this effect compared to simultaneous updates.

In Section 5.1, we present observations from our continuous-time models of momentum methods that support this thesis. Section 5.2 provides an experimental validation of this thesis. Additional numerical results are provided in Appendix J.

### 5.1. Observations from Continuous-time Models

To justify the above thesis, we first examine Continuous Sim-HB:

$$\dot{x} = \left(\frac{1}{\beta - 1}\right)\nabla_x f - \frac{h(1+\beta)}{4(1-\beta)^3}\nabla_x\left(\|\nabla_x f\|^2 - \|\nabla_y f\|^2\right)$$

$$\dot{y} = \left(\frac{1}{1 - \beta}\right)\nabla_y f - \frac{h(1+\beta)}{4(1-\beta)^3}\nabla_y\left(\|\nabla_y f\|^2 - \|\nabla_x f\|^2\right)$$

The evolution of the slope $\|\nabla_x f\|^2$ in these equations is influenced by two key terms. The first equation includes a gradient descent term for $\|\nabla_x f\|^2$ with respect to $x$:

$$-\frac{h(1+\beta)}{4(1-\beta)^3}\nabla_x\|\nabla_x f\|^2 = -\frac{h(1+\beta)}{4(1-\beta)^3}\nabla_x^2 f \cdot \nabla_x f, \quad (5)$$

which drives $\|\nabla_x f\|^2$ to decrease. Conversely, the second equation contains a gradient ascent term with respect to $y$:

$$\frac{h(1+\beta)}{4(1-\beta)^3}\nabla_y\|\nabla_x f\|^2 = \frac{h(1+\beta)}{4(1-\beta)^3}\nabla_{yx}f \cdot \nabla_x f, \quad (6)$$

which tends to increase $\|\nabla_x f\|^2$. Since the coefficient $h(1+\beta)/4(1-\beta)^3$ is an increasing function of $\beta$, a smaller $\beta$ reduces both effects.

In games satisfying Assumption 4.4, the interaction between players dominates game dynamics, that is, $\nabla_x^2 f(x,y) \ll \nabla_{yx}f(x,y)$. Thus, with a smaller momentum parameter $\beta$, the reduction in the gradient ascent effect from (6) outweighs the diminished descent effect from (5). Consequently, smaller $\beta$ leads to trajectories in Continuous Sim-HB with a shallower slope in terms of $\|\nabla_x f\|^2$. A similar conclusion applies to $\|\nabla_y f\|^2$.

Next, we consider Continuous Alt-HB. Since the $x$-player's equation remains the same as in simultaneous updates, we focus on the $y$-player's equation:

$$\dot{y} = \left(\frac{1}{1-\beta}\right)\nabla_y f - \frac{h(1+\beta)}{4(1-\beta)^3}\nabla_y\|\nabla_y f\|^2$$
$$+ \frac{h(3\beta - 1)}{4(1-\beta)^3}\nabla_y\|\nabla_x f\|^2 \quad (7)$$

The key difference between the $y$-player's equation in alternating and simultaneous updates lies in the coefficient of the $\nabla_y\|\nabla_x f\|^2$ term. In simultaneous updates, this coefficient is $h(1+\beta)/4(1-\beta)^3$, which is always positive for $\beta \in (-1, 1)$. However, in alternating updates, the coefficient changes to $h(3\beta - 1)/4(1-\beta)^3$, which becomes *negative* when $\beta \in (-1, \frac{1}{3})$. This implies that for small values of $\beta$, the $y$-player performs gradient descent on $\|\nabla_x f\|^2$, in order to minimize it. This is in contrast to simultaneous updates, where the $y$-player always maximizes $\|\nabla_x f\|^2$. Consequently, alternating updates encourage trajectories to explore regions with shallower slopes.

### 5.2. Experimental Results

Unlike minimization, where algorithms almost always converge to a local minimum (Lee et al., 2019), the limit sets of algorithms in min-max games are more complex. Beyond convergence to local equilibrium, typical behaviors include convergence to limit cycles and other non-convergent dynamics, commonly observed in practical tasks such as GAN training (Mescheder et al., 2018). Our experiments show that the thesis holds across these diverse dynamical patterns. Additional experiments are provided in Appendix J.

**Experiments on 2D test functions.** We present experiments on two 2D test functions, which lead the algorithms' trajectories converge to either a local equilibrium or a limit

cycle, and most parts of these trajectories satisfy Assumption 4.4. The test functions are described as follows:

*Test function 1.* The test function is $f(x, y) = -xy^2, x \geq 0$, where all points on the line $(x, 0)$ are equilibrium points. As shown in Figure 3(a), regions near equilibrium with smaller $x$ values have lower gradient norms.

*Test function 2.* The test function is $f(x, y) = 3x(4y - 0.45) + g(x) - g(y)$, where $g(z) = \frac{1}{2}z^2 - \frac{1}{2}z^4 + \frac{1}{6}z^6$. This construction was also used by (Hsieh et al., 2021; Pethick et al., 2023) to show that certain min-max algorithms exhibit limit cycles. As shown in Figure 3(c), the game has a unique equilibrium, and regions near the equilibrium exhibit lower gradient norms. Furthermore, under this test function, heavy ball momentum converges to limit cycles, highlighted by the white curves in Figure 3(c).

For each function, we present sample trajectories and the evolution of trajectories' average slopes as $\beta$ changes. The average slopes are numerically calculated through

$$\text{AvgSlope}(\beta) = \frac{1}{|S(\beta)|} \int_{S(\beta)} \left( \|\nabla_x f\|^2 + \|\nabla_y f\|^2 \right) ds$$

where $|S(\beta)|$ denotes the total trajectory length for a given momentum $S(\beta)$. *Lower* average slope values indicate that algorithms' trajectories explore *shallower* slope regions of min-max loss landscapes.

Figure 3(a) and 3(c) show that smaller $\beta$ guides the sample trajectories toward regions with lower slopes. Figure 3(b) and 3(d) demonstrate that for both simultaneous and alternating updates, smaller $\beta$ results in trajectories with lower average slopes. Additionally, for a fixed $\beta$, Alt-HB produces trajectories with lower average slopes than Sim-HB. These experimental results fully support our thesis.

**Experiments on GANs training.** We provide experiments on GANs training dynamics. The aim of these experiments is not to beat the state-of-the-art, but rather to validate our thesis on the implicit gradient regularization effect of momentum in GANs training setting. Future details and experiments are presented in Appendix J.2. Our setup generally follows the Wasserstein GANs framework (Gulrajani et al., 2017) using the CIFAR-10 dataset. We train GANs using the Adam algorithm with different heavy ball momentum parameters, updating the generator and discriminator either alternately or simultaneously. Future details include: Neural network architecture: Both generator and discriminator use the ResNet-32 architecture. Both generator and discriminator use the learning rate 2e-4, with a linearly decreasing step size schedule. The batch size is 64. During the training, we update both the generator and discriminator in each iteration, which is consistent with the algorithms investigated in this work.

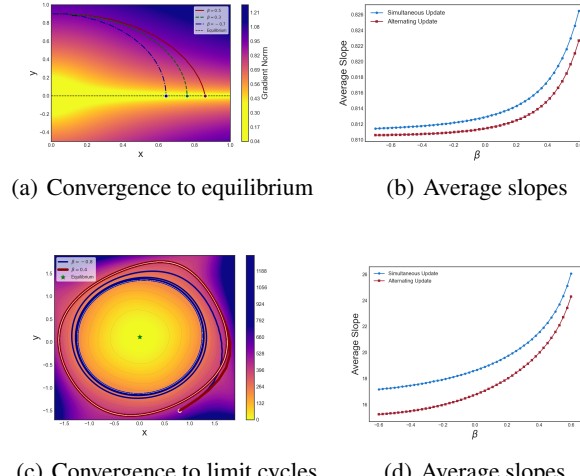

(a) Convergence to equilibrium

(b) Average slopes

(c) Convergence to limit cycles

(d) Average slopes

*Figure 3.* Trajectories and average slopes for test function 1 and 2. The background color in trajectories' pictures represent the magnitude of the gradient norm, i.e., $\|\nabla_x f\|^2 + \|\nabla_y f\|^2$.

To measure the implicit regularization effects, we evaluate the average slopes of algorithm trajectories, as in previous experiments. Since integrating over high-dimensional GANs training trajectories is challenging, we use cumulative average slopes as an alternative measure. At iteration $t$, the cumulative average slope of algorithms' trajectory is calculated as

$$\text{AvgS}_\beta(t) = \frac{1}{t} \sum_{s=1}^{t} \left( \|\nabla_x f(x_s, y_s)\|^2 + \|\nabla_y f(x_s, y_s)\|^2 \right).$$

Figure 4(a) shows the evolution of $\text{AvgS}_\beta(t)$ for trajectories with different $\beta$ under alternating updates. The results indicate that smaller momentum leads to smaller average slopes. Figure 4(b) compares two trajectories that differ only in their update rules, and alternating update has lower average slopes than simultaneous update. All of these results support the predictions made by the thesis.

Interestingly, the implicit gradient regularization effect appears to be linked to GAN training quality. In Figures 4(c) and 4(d), we use the Fréchet Inception Distance (FID) to evaluate GANs performance, where lower FID values indicate better GANs performance (Heusel et al., 2017). The results show that trajectories with lower average slopes correspond to better training outcomes. This may be attributed to the implicit gradient regularization effect, which guides optimization trajectories away from sharp gradient regions in the min-max loss landscape, and these regions are believed to be a major factor in GAN training failures (Thanh-Tung et al., 2019; Wu et al., 2021).

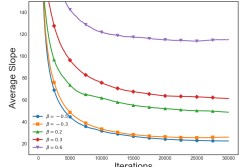
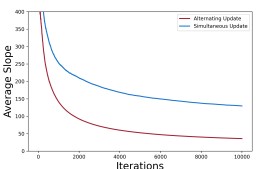

(a) Average Slopes, different $\beta$ (b) Average Slopes, alt vs. sim

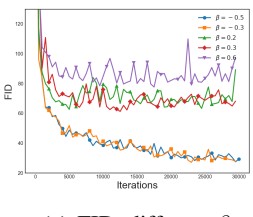
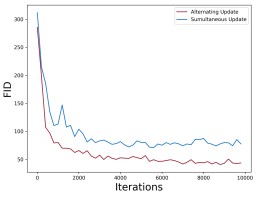

(c) FID, different $\beta$      (d) FID, alt vs. sim

*Figure 4.* Experimental results for GANs training dynamics. Smaller momentum and alternating updates lead the trajectories to lower average slopes. Trajectories with lower average slopes also have lower FID, indicating the better GANs training outcome.

# 6. Conclusions and Future Directions

In this paper, we explored the effects of momentum and updates rules in min-max games using continuous-time models. We analyzed the local convergence properties and the implicit gradient regularization effects of momentum algorithms, revealing fundamental differences in the role of momentum between min-max games and minimization. One of the limitations of this work is that we do not consider the potential impact of using stochastic gradient methods on algorithms' dynamics. Our results also suggest a potential link between implicit regularization and training outcomes in min-max games. Exploring this connection further and developing a general theoretical framework, similar to that established in the minimization setting, presents an exciting avenue for future research.

# Acknowledgments

The authors sincerely thank the valuable comments from the anonymous reviewers. Kaito Fujii was supported by FY2024 Researcher Exchange Program between JSPS and ETH and JSPS KAKENHI Grant Number 22K17857. Stratis Skoulakis is supported by Villum Young Investigator Award no. 72091. Xiao Wang acknowledges Grant 202110458 from SUFE and support from the Shanghai Research Center for Data Science and Decision Technology. This work was supported by Hasler Foundation Program: Hasler Responsible AI (project number 21043). Research was sponsored by the Army Research Office and was accomplished under Grant Number W911NF-24-1-0048. This work was supported by the Swiss National Science Foundation (SNSF) under grant number 200021_205011.

# Impact Statement

This paper presents work whose goal is to advance the field of Machine Learning. There are many potential societal consequences of our work, none of which we feel must be specifically highlighted here.

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

## A. More Experiments on Comparing Trajectories between Continuous-time Equations and Discrete-time Algorithms

In this section, we provide experiments in examples provided in Figure 1 of (Compagnoni et al., 2024b).

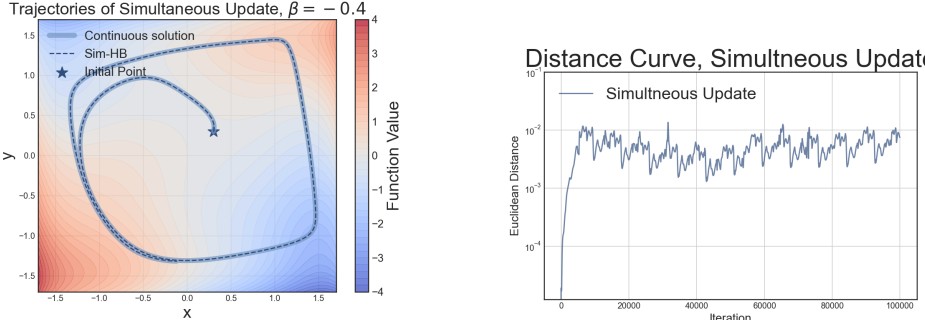

(a) Trajectories of Continuous Sim-HB and Sim-HB. (b) Distance between ODEs and algorithms, simultaneous update.

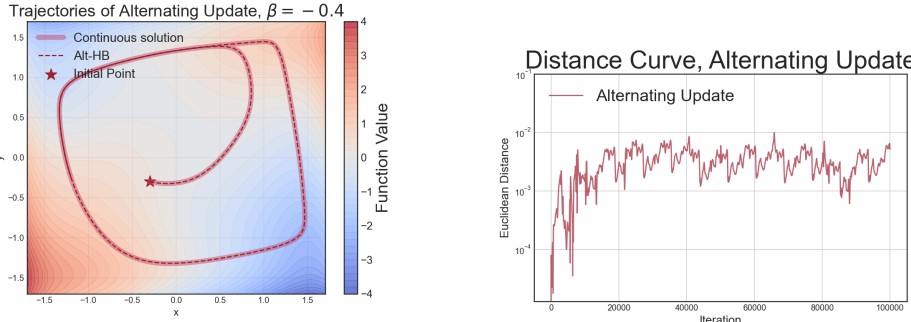

(c) Trajectories of Continuous Alt-HB and Alt-HB. (d) Distance between ODEs and algorithms, alternating update.

*Figure 5.* The test function is $f(x, y) = x(y - 0.45) + \phi(x) - \phi(y)$, $\phi(z) = \frac{1}{4}z^2 - \frac{1}{2}z^4 + \frac{1}{6}z^6$. This function is also used by **top left of Figure 1 in (Compagnoni et al., 2024)** to compare the trajectories between algorithms and their SDE models. Here we set step size $h = 0.001$ and $\beta = -0.4$ for the heavy ball method. The trajectories converge to limit cycles. From 5(a) and 5(c), the trajectories of the discrete-time algorithms closely match our continuous-time equations. In 5(b) and 5(d), we show the Euclidean distance between these trajectories. Specially, the distance between trajectories remains around 0.01 after 100000 iterations.

## B. Proof of Theorem 3.1

### B.1. Proof Overview and Discussions

There are several methods in the minimization literature to provide continuous-time equations to model the discrete-time momentum methods, for example, (Su et al., 2016; Shi et al., 2022) and (Muehlebach & Jordan, 2019). Most of these methods are specially designed for the setting where the objective function $f(x)$ is a (strongly) convex function, and the choice of step size and momentum are coupled together using numerical conditions of $f(x)$ such as the condition numbers. These methods are not suitable for our purposes because we aim to develop models that allow the momentum and step size to change independently, thus enabling us to better understand the effects of each term on the learning dynamics in min-max games.

The continuous-time models used in the current paper is inspired by recent work of (Ghosh et al., 2023), which try to use continuous-time models of heavy ball momentum to understand their implicit gradient regularization in minimization problems. The proof strategies of Theorem 3.1 can be summarized in two steps:

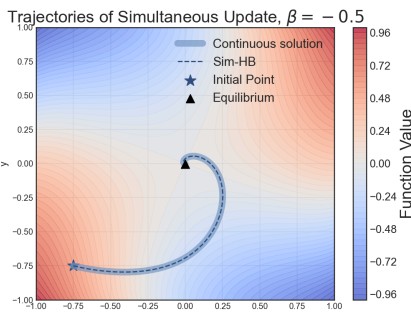
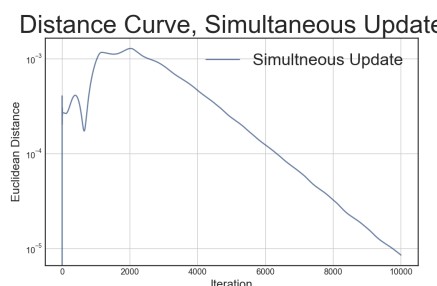

(a) Trajectories of Continuous Sim-HB and Sim-HB. (b) Distance between ODEs and algorithms, simultaneous update.

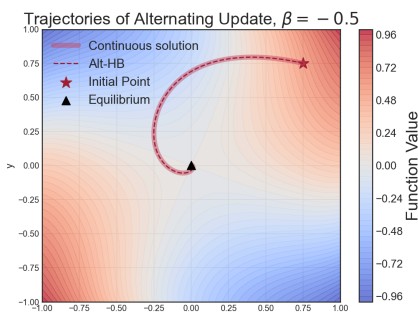
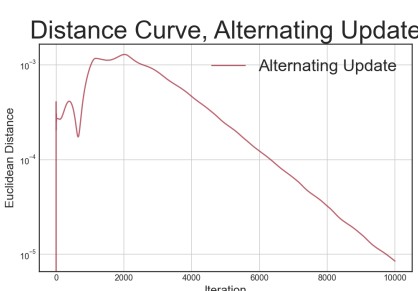

(c) Trajectories of Continuous Alt-HB and Alt-HB. (d) Distance between ODEs and algorithms, alternating update.

*Figure 6.* The test function is $f(x, y) = xy + \phi(x) - \phi(y)$, $\phi(z) = \frac{1}{2}z^2 - \frac{1}{4}z^4 + \frac{1}{6}z^6 - \frac{1}{8}z^8$. This function is also used by **bottom left of Figure 1 in (Compagnoni et al., 2024)** to compare the trajectories between algorithms and their SDE models. Here we set step size $h = 0.001$ and $\beta = -0.5$ for the heavy ball method. The trajectories converge to equilibrium. From 6(a) and 6(c), the trajectories of the discrete-time algorithms closely match our continuous-time equations. In 6(b) and 6(d), we show the Euclidean distance between these trajectories. We find these two curves only have **very slightly differences**. Moreover, the distances between these curves are close to 0.00001 after 10000 iterations
.

- Step 1: Firstly, we calculate a family of piecewise differential equations indexed by $n$, and each one is defined on the finite time region $[t_n, t_{n+1})$ where $t_n = hn$, and $h$ is the step size of the discrete-time algorithms. The solution of these equations can be linked to continuous curves that serve as good approximations of the original discrete-time algorithms. In our case, it can be proved that the local error between these continuous-time solutions and the discrete-time trajectories belongs to $\mathcal{O}(h^3)$. However, this family of equations has several drawbacks: they are not differentiable on the time nodes, and their formulation is dependent on $n$, which lead them to be complex to analysis.

- Step 2: Secondly, we show that this family of differential equations can converge to a differential equation which is independent of $n$ and is differentiable everywhere with an exponential rate. This exponential rate convergence enables us to consider the solution of this single differential equation instead of the family of equations defined in Step 1. In fact, we can prove that after a very short time region, the distance between the trajectories of this single equation and the solution of the equations defined in step 1 stops growth and can be bounded in $\mathcal{O}(h^3)$ local error. This provide the theoretical guarantee that this single equation can serve as a good approximation of the original discrete-time algorithms.

The above two steps are needed both for simultaneous updates and alternating updates of heavy ball momentum in min-max games. In Lemma B.1 and Lemma B.4, we will calculate the family of piecewise differentiable equations for Sim-HB and Alt-HB. We will see that because there are two players in a min-max game, the continuous-time equations we get for heavy

ball momentum methods are very different from the one in (Ghosh et al., 2023).

For example, if we consider the simultaneous update, as Lemma B.1 shows, the $x$-player's equation is written as

$$\dot{x}(t) = -\left(\frac{1-\beta^n}{1-\beta}\right)\nabla_x f(x(t),y(t)) - \frac{h\gamma_n}{2(1-\beta)^2}\left(\nabla_x^2 f(x(t),y(t))\cdot\nabla_x f(x(t),y(t)) - \nabla_{xy} f(x(t),y(t))\cdot\nabla_y f(x(t),y(t))\right),$$

and the equations in (Ghosh et al., 2023) (Theorem 4.1 in their paper) are written as

$$\dot{x}(t) = -\left(\frac{1-\beta^n}{1-\beta}\right)\nabla_x f(x(t)) - \frac{h\gamma_n}{2(1-\beta)^2}\nabla_x^2 f(x(t))\cdot\nabla_x f(x(t)),$$

the key term $\nabla_{xy} f(x(t),y(t))\cdot\nabla_y f(x(t),y(t))$ is important for understanding the great differences on the role of momentum in min-max games and minimization.

In Lemma B.3 and B.5, we prove the family of piecewise differentiable equations defined in Lemma B.1 and Lemma B.4 can quickly converge to equations independent of index $n$. This can be seen from the fact that the dependence of $n$ in Lemma B.1 and Lemma B.4 appears only in the form of $\beta^n$. Since $\beta \in (-1,1)$, $\beta^n$ vanishes with an exponential rate.

## B.2. Simultaneous Updates

**Lemma B.1.** *Let $t_n = nh$ where $h$ is the step size used by Sim-HB. Then in time region $[t_n, t_{n+1})$, the solution of the following ODEs:*

$$\dot{x}(t) = -\left(\frac{1-\beta^n}{1-\beta}\right)\nabla_x f(x(t),y(t))$$

$$-\frac{h\gamma_n}{2(1-\beta)^2}\left(\nabla_x^2 f(x(t),y(t))\cdot\nabla_x f(x(t),y(t)) - \nabla_{xy} f(x(t),y(t))\cdot\nabla_y f(x(t),y(t))\right)$$

$$\dot{y}(t) = \left(\frac{1-\beta^n}{1-\beta}\right)\nabla_y f(x(t),y(t))$$

$$+\frac{h\gamma_n}{2(1-\beta)^2}\left(\nabla_{yx} f(x(t),y(t))\cdot\nabla_x f(x(t),y(t)) - \nabla_y^2 f(x(t),y(t))\cdot\nabla_y f(x(t),y(t))\right),$$

*can approximate the trajectories of Sim-HB with an $\mathcal{O}(h^3)$-local error.*

*Here $\gamma_n = \sum_{i=0}^n \beta^{n-i}\left[(1+\beta)(1+\beta^{2i+1}) - 4\beta^{i+1}\right]$.*

*Proof.* Assume in time region $[t_n, t_{n+1})$, we have

$$\dot{x}(t) = -\nabla_x G_n(x(t),y(t)) + hA_n(x(t),y(t)) \tag{8}$$

$$\dot{y}(t) = \nabla_y F_n(x(t),y(t)) + hB_n(x(t),y(t)) \tag{9}$$

Our aim is to find functions $G_n(x(t),y(t))$, $F_n(x(t),y(t))$, $A_n(x(t),y(t))$, and $B_n(x(t),y(t))$ such that

$$(x(t_{n+1}) - x(t_n)) - \beta(x(t_n) - x(t_{n-1})) = -h\nabla_x f(x(t_n),y(t_n)) + \mathcal{O}(h^3) \tag{10}$$

$$(y(t_{n+1}) - y(t_n)) - \beta(y(t_n) - y(t_{n-1})) = h\nabla_y f(x(t_n),y(t_n)) + \mathcal{O}(h^3) \tag{11}$$

hold.

In the following, we will use the notation $\dot{x}(t_n^+)$ (resp. $\ddot{x}(t_n^+)$) to denote the right first-order derivative (resp. second-order derivative) of $x$ at $t_n$. Similarly, the notation $\dot{x}(t_n^-)$ (resp. $\ddot{x}(t_n^-)$) is used to denote the left first-order derivative (resp. second-order derivative) of $x$ at $t_n$.

From Lemma B.2, we have $\dot{x}(t_n^+), \ddot{x}(t_n^+)$, and $\dddot{x}(t_n^+)$ are bounded independent of $h$. Thus, by Taylor expansion, we have

$$x(t_{n+1}) - x(t_n) = x(t_n + h) - x(t_n)$$
$$= h\dot{x}(t_n^+) + \frac{h^2}{2}\ddot{x}(t_n^+) + \mathcal{O}(h^3), \tag{12}$$

From (8) and (9), we can calculate $\ddot{x}(t_n^+)$ as

$$\ddot{x}(t_n^+) = -\nabla_x^2 G_n(x(t_n), y(t_n)) \cdot \dot{x}(t_n^+) - \nabla_{xy} G_n(x(t_n), y(t_n)) \cdot \dot{y}(t_n^+) + \mathcal{O}(h)$$

$$= \nabla_x^2 G_n(x(t_n), y(t_n)) \cdot \nabla_x G_n(x(t_n), y(t_n)) - \nabla_{xy} G_n(x(t_n), y(t_n)) \cdot \nabla_y F_n(x(t_n), y(t_n)) + \mathcal{O}(h) \tag{13}$$

Take (8) and (13) into (12), we get

$$x(t_{n+1}) - x(t_n)$$
$$= -h\nabla_x G_n(x(t_n), y(t_n)) + h^2 A_n(x(t_n), y(t_n))$$
$$+ \frac{h^2}{2}\left(\nabla_x^2 G_n(x(t_n), y(t_n))\nabla_x G_n(x(t_n), y(t_n)) - \nabla_{xy} G_n(x(t_n), y(t_n))\nabla_y F_n(x(t_n), y(t_n))\right) + \mathcal{O}(h^3) \tag{14}$$

With a similar calculation on the Taylor expansion of $x(t_{n-1})$ at point $x(t_n)$, we get

$$-\beta\left(x(t_n) - x(t_{n-1})\right) = -\beta\left(h\dot{x}(t_n^-) - \frac{h^2}{2}\ddot{x}(t_n^-)\right)$$
$$= \beta h \nabla_x G_{n-1}(x(t_n), y(t_n))$$
$$- \beta h^2 [A_{n-1}(x(t_n), y(t_n))$$
$$- \frac{1}{2}(\nabla_x^2 G_{n-1}(x(t_n), y(t_n))\nabla_x G_{n-1}(x(t_n), y(t_n)) - \nabla_{xy} G_{n-1}(x(t_n), y(t_n))\nabla_y F_{n-1}(x(t_n), y(t_n)))]$$
$$+ \mathcal{O}(h^3) \tag{15}$$

Recall that our aim is to determine the formulation of $G_n(x, y), F_n(x, y), A_n(x, y), B_n(x, y)$ such that (10) holds. From (14) and (15), we have calculated the Taylor expansion of $x(t_{n+1}) - x(t_n) - \beta\left(x(t_n) - x(t_{n-1})\right)$.

Next, we will compare the coefficients of this Taylor expansion with (10), which gives us

$$-h\left(\nabla_x G_n(x(t_n), y(t_n)) - \beta \nabla_x G_{n-1}(x(t_n), y(t_n))\right) = -h\nabla_x f\left(x(t_n), y(t_n)\right), \tag{16}$$

and

$$A_n(x(t_n), y(t_n)) + \frac{1}{2}\left(\nabla_x^2 G_n(x(t_n), y(t_n)) \cdot \nabla_x G_n(x(t_n), y(t_n)) - \nabla_{xy} G_n(x(t_n), y(t_n)) \cdot \nabla_y F_n(x(t_n), y(t_n))\right)$$

$$- \beta[A_{n-1}(x(t_n), y(t_n)) - \frac{1}{2}(\nabla_x^2 G_{n-1}(x(t_n), y(t_n)) \cdot \nabla_x G_{n-1}(x(t_n), y(t_n))$$
$$- \nabla_{xy} G_{n-1}(x(t_n), y(t_n)) \cdot \nabla_y F_{n-1}(x(t_n), y(t_n)))]$$
$$= 0. \tag{17}$$

Note that (16) gives a recurrence relation between $\nabla_x G_n\left(x(t_n), y(t_n)\right)$ and $\nabla_x G_{n-1}\left(x(t_n), y(t_n)\right)$ for $n \in \mathbb{N}$, and solving this relation gives

$$\nabla_x G_n\left(x(t_n), y(t_n)\right) = \frac{1 - \beta^{n+1}}{1 - \beta}\nabla_x f\left(x(t_n), y(t_n)\right). \tag{18}$$

With a similar argument for $y$-player's equation (9), we can get

$$\nabla_y F_n\left(x(t_n), y(t_n)\right) = \frac{1 - \beta^{n+1}}{1 - \beta}\nabla_y f\left(x(t_n), y(t_n)\right). \tag{19}$$

Next we solve $A_n(x, y)$ from (17). We first rewrite (17) as the following form:

$$
\begin{aligned}
& A_n\left(x(t_n), y(t_n)\right) - \beta A_{n-1}\left(x(t_n), y(t_n)\right) \\
& = -\frac{1}{2}(\nabla_x^2 G_n\left(x(t_n), y(t_n)\right) \cdot \nabla_x G_n\left(x(t_n), y(t_n)\right) + \beta \nabla_x^2 G_{n-1}\left(x(t_n), y(t_n)\right) \cdot \nabla_x G_{n-1}\left(x(t_n), y(t_n)\right) \\
& \quad - (\nabla_{xy} G_n\left(x(t_n), y(t_n)\right) \cdot \nabla_y F_n\left(x(t_n), y(t_n)\right) + \beta \nabla_{xy} G_{n-1}\left(x(t_n), y(t_n)\right) \cdot \nabla_y F_{n-1}\left(x(t_n), y(t_n)\right)))
\end{aligned} \tag{20}
$$

From (18) and (19), we have

$$
\begin{aligned}
& \nabla_x^2 G_n\left(x(t_n), y(t_n)\right) \cdot \nabla_x G_n\left(x(t_n), y(t_n)\right) + \beta \nabla_x^2 G_{n-1}\left(x(t_n), y(t_n)\right) \cdot \nabla_x G_{n-1}\left(x(t_n), y(t_n)\right) \\
& = \left(\left(\frac{1 - \beta^{n+1}}{1 - \beta}\right)^2 + \beta\left(\frac{1 - \beta^n}{1 - \beta}\right)^2\right) \nabla_x^2 f\left(x(t_n), y(t_n)\right) \cdot \nabla_x f\left(x(t_n), y(t_n)\right)
\end{aligned} \tag{21}
$$

and

$$
\begin{aligned}
& \nabla_{xy} G_n\left(x(t_n), y(t_n)\right) \cdot \nabla_y F_n\left(x(t_n), y(t_n)\right) + \beta \nabla_{xy} G_{n-1}\left(x(t_n), y(t_n)\right) \cdot \nabla_y F_{n-1}\left(x(t_n), y(t_n)\right) \\
& = \left(\left(\frac{1 - \beta^{n+1}}{1 - \beta}\right)^2 + \beta\left(\frac{1 - \beta^n}{1 - \beta}\right)^2\right) \nabla_{xy} f\left(x(t_n), y(t_n)\right) \cdot \nabla_y f\left(x(t_n), y(t_n)\right)
\end{aligned} \tag{22}
$$

Taking (21) and (22) into (20), we obtain a recurrence relation between $A_n\left(x(t_n), y(t_n)\right)$ and $A_{n-1}\left(x(t_n), y(t_n)\right)$. By solving this recurrence relation, we can derive the formulation of $A_n\left(x(t_n), y(t_n)\right)$:

$$
\begin{aligned}
& A_n\left(x(t_n), y(t_n)\right) \\
& = -\frac{1}{2(1 - \beta)^2} \gamma_n (\nabla_x^2 f\left(x(t_n), y(t_n)\right) \cdot \nabla_x f\left(x(t_n), y(t_n)\right) - \nabla_{xy} f\left(x(t_n), y(t_n)\right) \cdot \nabla_y f\left(x(t_n), y(t_n)\right)),
\end{aligned} \tag{23}
$$

where $\gamma_n = \sum_{i=0}^{n} \beta^{n-i}\left[(1 + \beta)(1 + \beta^{2i+1}) - 4\beta^{i+1}\right]$.

Combining the above, we have determined the formulation of $\dot{x}$ in time region $[t_n, t_{n+1})$ as follows:

$$
\begin{aligned}
\dot{x}(t) = {} & \left(\frac{1 - \beta^{n+1}}{1 - \beta}\right) \nabla_x f\left(x(t_n), y(t_n)\right) \\
& - \frac{h}{2(1 - \beta)^2} \gamma_n \left(\nabla_x^2 f\left(x(t_n), y(t_n)\right) \cdot \nabla_x f\left(x(t_n), y(t_n)\right) - \nabla_{xy} f\left(x(t_n), y(t_n)\right) \cdot \nabla_y f\left(x(t_n), y(t_n)\right)\right).
\end{aligned} \tag{24}
$$

This completes the proof of the part of Lemma B.1. Since in Simultaneous update, $x(t)$ and $y(t)$ are symmetrical, the part for $y(t)$ is similar to the above, and we omit it here. $\qquad \square$

**Lemma B.2.** *Under the assumption that $f(x, y)$ is a smooth function with bounded derivatives up to the fourth order as stated in Theorem 3.1, then $\dot{x}(t), \ddot{x}(t)$ and $\dddot{x}(t)$ are bounded independent of $h$.*

*Proof.* By assumption, we have $\|\nabla_{a,b,c,d} f(x, y)\|$ are bounded independent of $h$, where $a, b, c, d$ equal to $x$ or $y$. We also assume the derivatives of $f(x, y)$ up to the fourth order are bounded by a constant $\mathcal{C}$.

The aim of this Lemma is to guarantee that we can faithfully truncate the Taylor-expansion in (12) to get an error of $\mathcal{O}(h^3)$. To achieve this, we need to make sure that the terms $\dot{x}(t), \ddot{x}(t)$ and $\dddot{x}(t)$ are bounded independent of $h$.

Let firstly consider $\dot{x}(t)$. From (24) and the boundedness assumption of the first- and second-order derivatives, we have

$$
\begin{aligned}
\|\dot{x}(t)\| \leq {} & \left(\frac{1 - \beta^{n+1}}{1 - \beta}\right) \|\nabla_x f\left(x(t_n), y(t_n)\right)\| + \frac{h}{2(1 - \beta)^2} \gamma_n \|\nabla_x^2 f\left(x(t_n), y(t_n)\right)\| \cdot \|\nabla_x f\left(x(t_n), y(t_n)\right)\| \\
& + \frac{h}{2(1 - \beta)^2} \gamma_n \|\nabla_{xy} f\left(x(t_n), y(t_n)\right)\| \cdot \|\nabla_y f\left(x(t_n), y(t_n)\right)\| \\
& \leq \left(\frac{1 - \beta^{n+1}}{1 - \beta}\right) \cdot \mathcal{C} + \frac{h}{(1 - \beta)^2} \gamma_n \mathcal{C}^2 \leq \mathcal{C}_2.
\end{aligned}
$$

Here we use the boundedness of $h$, which is a small positive number, and $\mathcal{C}_2$ is a constant number independent of $h$. Note that from (12), $\dddot{x}(t)$ and $\ddddot{x}(t)$ can also be formulated as combinations of the derivatives of $f(x,y)$ up to the fourth order. Therefore, their boundedness can be proved in a similar way. $\qquad\square$

**Lemma B.3.** *Let $\mathcal{T}$ be a fixed time horizon, and $t \in [0, \mathcal{T}]$. Let $(\tilde{x}(t), \tilde{y}(t))$ denote the solution trajectory of the equations defined in Lemma B.1, and $(x(t), y(t))$ denote the solution trajectory of the equations defined in Theorem 3.1 for Continuous Sim-HB. If $n > 2\log(h)/\log(\beta)$, and at time $t_n = nh$ we have*

$$(\tilde{x}(t_n), \tilde{y}(t_n)) = (x(t_n), y(t_n)),$$

*then at time $t_{n+1} = (n+1)h$, we have*

$$\| (x(t_{n+1}), y(t_{n+1})) - (\tilde{x}(t_{n+1}), \tilde{y}(t_{n+1})) \| = \mathcal{O}(h^3).$$

*Proof.* We will prove

$$\|x(t_{n+1}) - \tilde{x}(t_{n+1})\| = \mathcal{O}(h^3),$$

and the argument for $y$-player follows similarly.

Using Taylor expansion, we have

$$\tilde{x}(nh + h) - \tilde{x}(nh) = h\dot{\tilde{x}}(nh^+) + \frac{h^2}{2}\ddot{\tilde{x}}(nh^+) + \mathcal{O}(h^3),$$

$$x(nh + h) - x(nh) = h\dot{x}(nh) + \frac{h^2}{2}\ddot{x}(nh) + \mathcal{O}(h^3).$$

Thus, we obtain

$$\|x(t_{n+1}) - \tilde{x}(t_{n+1})\| = h\|(\dot{\tilde{x}}(nh^+) - \dot{x}(nh))\| + \frac{h^2}{2}\|(\ddot{\tilde{x}}(nh^+) - \ddot{x}(nh))\| + \mathcal{O}(h^3).$$

In the following, we will show both of the terms $h\|(\dot{\tilde{x}}(nh^+) - \dot{x}(nh))\|$ and $\frac{h^2}{2}\|(\ddot{\tilde{x}}(nh^+) - \ddot{x}(nh))\|$ belong to $\mathcal{O}(h^3)$.

From Continuous Sim-HB, we have

$$h\dot{x}(nh) = -\left(\frac{h}{1-\beta}\right)\nabla_x f(x(nh), y(nh)) - \frac{h^2(1+\beta)}{2(1-\beta)^3}(\nabla_x^2 f(x(nh), y(nh)) \cdot \nabla_x f(x(nh), y(nh))$$

$$- \nabla_{xy} f(x(nh), y(nh)) \cdot \nabla_y f(x(nh), y(nh))). \qquad (25)$$

From Lemma B.1, we have

$$h\dot{\tilde{x}}(nh) = -\frac{h(1-\beta^n)}{1-\beta}\nabla_x f(x(nh), y(nh))$$

$$- \frac{h^2(1+\beta)}{2(1-\beta)^3}\left[(1-\beta^{2n+2}) - 4(n+1)\beta^{n+1}\left(\frac{1-\beta}{1+\beta}\right)\right] \cdot [\nabla_x^2 f(x(nh), y(nh)) \cdot \nabla_x f(x(nh), y(nh))$$

$$- \nabla_{xy} f(x(nh), y(nh)) \cdot \nabla_y f(x(nh), y(nh))]. \qquad (26)$$

Using (25) minus (26), we get

$$h\dot{x}(nh) - h\dot{\tilde{x}}(nh) = -\frac{h}{1-\beta}\beta^n\nabla_x f(x(nh), y(nh)) - \frac{h^2(1+\beta)}{2(1-\beta)^3}\left[\beta^{2n+2} + 4(n+1)\beta^{n+1}\left(\frac{1-\beta}{1+\beta}\right)\right] \cdot$$

$$\left(\nabla_x^2 f(x(nh), y(nh)) \cdot \nabla_x f(x(nh), y(nh)) - \nabla_{xy} f(x(nh), y(nh)) \cdot \nabla_y f(x(nh), y(nh))\right). \qquad (27)$$

Since $n > 2\log(h)/\log(\beta) \Leftrightarrow \beta^n < h^2$ and $f(x,y)$ have bounded first- and second-order derivatives, we have the first term in the above equality, i.e., $-\frac{h}{1-\beta}\beta^n\nabla_x f$, belongs to $\mathcal{O}(h^3)$.

Similarly, for the term

$$\frac{h^2(1+\beta)}{2(1-\beta)^3}\left[\beta^{2n+2} + 4(n+1)\beta^{n+1}\left(\frac{1-\beta}{1+\beta}\right)\right] \cdot$$
$$\left(\nabla_x^2 f(x(nh), y(nh)) \cdot \nabla_x f(x(nh), y(nh)) - \nabla_{xy} f(x(nh), y(nh)) \cdot \nabla_y f(x(nh), y(nh))\right)$$

in $h\dot{x}(nh) - h\dot{\tilde{x}}(nh)$, due to the boundedness of first- and second-order derivatives, we only need to show that the coefficient

$$\frac{h^2(1+\beta)}{2(1-\beta)^3}\left[\beta^{2n+2} + 4(n+1)\beta^{n+1}\left(\frac{1-\beta}{1+\beta}\right)\right]$$

belongs to $\mathcal{O}(h^3)$. This can be guaranteed since we have $n > 2\log(h)/\log(\beta) \Leftrightarrow \beta^n < h^2$, and this implies

$$(n+1)\beta^{n+1} < \frac{\mathcal{T}}{h}\beta^{n+1} < \mathcal{T}\beta h, \quad \beta^{2n+2} < h^2\beta^2.$$

Similarly, under the same condition $n > 2\log(h)/\log(\beta)$, we can prove $h^2\ddot{x}(nh) - h^2\ddot{\tilde{x}}(nh) \in \mathcal{O}(h^3)$ given the condition that $f(x,y)$ has bounded derivatives up to the third order. $\qquad\square$

## B.3. Alternating Updates

**Lemma B.4.** *Let $t_n = nh$ where $h$ is the step size used by* Alt-HB. *Then in time region $[t_n, t_{n+1})$, the solution of the following ODEs:*

$$\dot{x}(t) = -\left(\frac{1-\beta^n}{1-\beta}\right)\nabla_x f(x(t), y(t))$$
$$\quad - \frac{h\gamma_n}{2(1-\beta)^2}\left(\nabla_x^2 f(x(t), y(t)) \cdot \nabla_x f(x(t), y(t)) - \nabla_{xy} f(x(t), y(t)) \cdot \nabla_y f(x(t), y(t))\right),$$
$$\dot{y}(t) = \left(\frac{1-\beta^n}{1-\beta}\right)\nabla_y f(x(t), y(t))$$
$$\quad + \frac{h\gamma_n}{2(1-\beta)^2}\left(\nabla_{yx} f(x(t), y(t)) \cdot \nabla_x f(x(t), y(t)) - \nabla_y^2 f(x(t), y(t)) \cdot \nabla_y f(x(t), y(t))\right)$$
$$\quad - \frac{h\delta_n}{2(1-\beta)^2}\nabla_{yx} f(x(t), y(t)) \cdot \nabla_x f(x(t), y(t)).$$

*can approximate the trajectories of* Alt-HB *with a $\mathcal{O}(h^3)$-local error. Here $\gamma_n = \sum_{i=0}^n \beta^{n-i}\left[(1+\beta)(1+\beta^{2i+1}) - 4\beta^{i+1}\right]$, and $\delta_n = 2\sum_{i=0}^n \beta^{n-i}(1-\beta^{i+1})(1-\beta)$.*

*Proof.* Recall that Alt-HB is written as

$$x_{n+1} = x_n - h\nabla_x f(x_n, y_n) + \beta(x_n - x_{n-1}) \tag{28}$$
$$y_{n+1} = y_n + h\nabla_x f(x_{n+1}, y_n) + \beta(y_n - y_{n-1}) \tag{29}$$

Assume in time region $[t_n, t_{n+1})$, we have

$$\dot{x}(t) = -\nabla_x G_n(x, y) + hA_n(x, y) \tag{30}$$
$$\dot{y}(t) = \nabla_y F_n(x, y) + hB_n(x, y) \tag{31}$$

Our aim is to find functions $G_n(x,y)$, $F_n(x,y)$, $A_n(x,y)$, and $B_n(x,y)$ such that

$$(x(t_{n+1}) - x(t_n)) - \beta(x(t_n) - x(t_{n-1})) = -h\nabla_x f(x(t_n), y(t_n)) + \mathcal{O}(h^3) \tag{32}$$

and

$$(y(t_{n+1}) - y(t_n)) - \beta(y(t_n) - y(t_{n-1})) = h\nabla_y f(x(t_{n+1}), y(t_n)) + \mathcal{O}(h^3) \tag{33}$$

hold.

In this proof, we will focus on the derivation of $F_n(x, y)$ and $B_n(x, y)$, since the derivation of $G_n(x, y)$ and $A_n(x, y)$ is the same as in Lemma B.1.

In the following, we will use the notation $\dot{y}(t_n^+)$ (resp. $\ddot{y}(t_n^+)$) to denote the right first-order derivative (resp. second-order derivative) of $x$ at $t_n$. Similarly, the notation $\dot{y}(t_n^-)$ (resp. $\ddot{y}(t_n^-)$) is used to denote the left first-order derivative (resp. second-order derivative) of $y$ at $t_n$.

We begin by performing a Taylor expansion of $h\nabla_y f(x(t_{n+1}), y(t_n))$ at time $t_n$. Recall that $t_n = nh$, then

$$h\nabla_y f(x(t_n + h), y(t_n)) = h\nabla_y f(x(t_n), y(t_n)) + h^2 \nabla_{yx} f(x(t_n), y(t_n)) \cdot \dot{x}(t_n^+) + \mathcal{O}(h^3)$$
$$= h\nabla_y f(x(t_n), y(t_n)) - h^2 \left(\frac{1 - \beta^{n+1}}{1 - \beta}\right) \nabla_{yx} f(x(t_n), y(t_n)) \nabla_x f(x(t_n), y(t_n)) + \mathcal{O}(h^3). \tag{34}$$

Note that in (34), we use the fact that

$$\dot{x}(t_n^+) = -\left(\frac{1 - \beta^{n+1}}{1 - \beta}\right) \nabla_x f(x(t_n), y(t_n)) + \mathcal{O}(h),$$

which can be derived in the same way as in Lemma B.1.

Using the Taylor expansion of $y(t_n + h)$ at time $t_n$, we also have

$$y(t_n + h) = y(t_n) + h\dot{y}(t_n^+) + \frac{h^2}{2}\ddot{y}(t_n^+) + \mathcal{O}(h^3). \tag{35}$$

From (31), we have

$$\dot{y}(t_n^+) = \nabla_y F_n(x(t_n), y(t_n)) + hB_n(x(t_n), y(t_n)) \tag{36}$$

and

$$\ddot{y}(t_n^+) = \nabla_y^2 F_n(x(t_n), y(t_n)) \cdot \dot{y}(t_n^+) + \nabla_{yx} F_n(x(t_n), y(t_n)) \cdot \dot{x}(t_n^+) + \mathcal{O}(h)$$
$$= \nabla_y^2 F_n(x(t_n), y(t_n)) \cdot \nabla_y F_n(x(t_n), y(t_n)) - \nabla_{yx} F_n(x(t_n), y(t_n)) \cdot \nabla_x G_n(x(t_n), y(t_n)) + \mathcal{O}(h) \tag{37}$$

Taking above into (35), we can get the following formula of $y(t_{n+1}) - y(t_n)$ :

$$y(t_n + h) - y(t_n) = h\dot{y}(t_n^+) + \frac{h^2}{2}\ddot{y}(t_n^+) + \mathcal{O}(h^3)$$
$$= h\nabla_y F_n(x(t_n), y(t_n)) + h^2 B_n(x(t_n), y(t_n))$$
$$+ \frac{h^2}{2}\left(\nabla_y^2 F_n(x(t_n), y(t_n)) \cdot \nabla_y F_n(x(t_n), y(t_n)) - \nabla_{yx} F_n(x(t_n), y(t_n)) \cdot \nabla_x G_n(x(t_n), y(t_n))\right) + \mathcal{O}(h^3). \tag{38}$$

Similarly, we can get the Taylor expansion of $y(t_n - h)$ as time $t_n$. Note that here we will use the left derivatives rather than the right derivatives as in (38). The calculation will be omitted, and we only present the final results here:

$$\beta\left(y(t_n) - y(t_n - h)\right) = \beta h\nabla_y F_{n-1}(x(t_n), y(t_n))$$
$$+ \beta h^2 [B_{n-1}(x(t_n), y(t_n)) - \frac{1}{2}(\nabla_y^2 F_{n-1}(x(t_n), y(t_n)) \cdot \nabla_y F_{n-1}(x(t_n), y(t_n))$$
$$- \nabla_{yx} F_{n-1}(x(t_n), y(t_n)) \cdot \nabla_x G_{n-1}(x(t_n), y(t_n)))]. \tag{39}$$

Combine (39) and (38), we get

$$
\begin{aligned}
&(y(t_{n+1}) - y(t_n)) - \beta \left(y(t_n) - y(t_{n-1})\right) \\
&= h \left[\nabla_y F_n(x(t_n), y(t_n)) - \beta \nabla_y F_{n-1}(x(t_n), y(t_n))\right] \\
&+ h^2 \left[B_n(x(t_n), y(t_n)) - \beta B_{n-1}(x(t_n), y(t_n))\right] \\
&+ \frac{h^2}{2} \left[\nabla_y^2 F_n(x(t_n), y(t_n)) \cdot \nabla_y F_n(x(t_n), y(t_n)) + \beta \nabla_y^2 F_{n-1}(x(t_n), y(t_n)) \cdot \nabla_y F_{n-1}(x(t_n), y(t_n))\right] \\
&- \frac{h^2}{2} \left[\nabla_{yx} F_n(x(t_n), y(t_n)) \cdot \nabla_x G_n(x(t_n), y(t_n)) + \beta \nabla_{yx} F_{n-1}(x(t_n), y(t_n)) \cdot \nabla_x G_{n-1}(x(t_n), y(t_n))\right] \qquad (40) \\
&+ \mathcal{O}(h^3). \qquad (41)
\end{aligned}
$$

Now we compare (34) and (41), we get the following two equalities:

$$
\nabla_y F_n(x_n, y_n) - \beta \nabla_y F_{n-1}(x_n, y_n) = \nabla_y f(x_n, y_n) \qquad (42)
$$

and

$$
\begin{aligned}
&[B_n(x_n, y_n) - \beta B_{n-1}(x_n, y_n)] \\
&+ \frac{1}{2} \left[\nabla_y^2 F_n(x(t_n), y(t_n)) \cdot \nabla_y F_n(x(t_n), y(t_n)) + \beta \nabla_y^2 F_{n-1}(x(t_n), y(t_n)) \cdot \nabla_y F_{n-1}(x(t_n), y(t_n))\right] \\
&- \frac{1}{2} \left[\nabla_{yx} F_n(x(t_n), y(t_n)) \cdot \nabla_x G_n(x(t_n), y(t_n)) + \beta \nabla_{yx} F_{n-1}(x(t_n), y(t_n)) \cdot \nabla_x G_{n-1}(x(t_n), y(t_n))\right] \\
&= -\frac{1 - \beta^{n+1}}{1 - \beta} \nabla_{yx} f(x_n, y_n) \cdot \nabla_x f(x_n, y_n). \qquad (43)
\end{aligned}
$$

(42) gives a recurrence relation between $F_n$ and $F_{n-1}$, and solving this relation gives

$$
\nabla_y F_n(x, y) = \frac{1 - \beta^{n+1}}{1 - \beta} \nabla_y f(x, y). \qquad (44)
$$

As we have also known

$$
\nabla_x G_n(x, y) = \frac{1 - \beta^{n+1}}{1 - \beta} \nabla_x f(x, y), \qquad (45)
$$

taking $\nabla_y F_n(x, y), \nabla_x G_n(x, y)$ into (43), we can get

$$
B_n(x, y) = -\frac{1 + \beta}{2(1 - \beta)^3} \nabla_y^2 f(x, y) \cdot \nabla_y f(x, y) + \frac{3\beta - 1}{2(1 - \beta)^3} \nabla_{yx} f(x, y) \cdot \nabla_x f(x, y), \qquad (46)
$$

and this finished the proof. $\qquad \square$

**Lemma B.5.** *Let $(\tilde{x}(t), \tilde{y}(t))$ denote the solution trajectory of the equations defined in Lemma B.4, and $(x(t), y(t))$ denotes the solution trajectory of the equations defined in Theorem 3.1 for* Continuous Alt-HB. *If $n > 2 \log(h)/\log(\beta)$, then at time $t_n = nh$ we have*

$$
(\tilde{x}(t_n), \tilde{y}(t_n)) = (x(t_n), y(t_n)),
$$

*then at time $t_{n+1} = (n+1)h$, we have*

$$
\|(x(t_{n+1}), y(t_{n+1})) - (\tilde{x}(t_{n+1}), \tilde{y}(t_{n+1}))\| = \mathcal{O}(h^3).
$$

The proof of this lemma follows in the same way as that of Lemma B.3; thus, we omit it here.

## C. Proof of Proposition 3.2

The proof of Proposition 3.2 is divided into two parts. In Lemma C.1, we prove the part for Continuous Sim-HB. In Lemma C.2, we prove the part for Continuous Alt-HB. Proposition 3.2 directly follows from these two lemmas.

**Lemma C.1.** *In matrix games $f(x,y) = x^\top A y$, Continuous Sim-HB diverges at an exponential rate regardless of the choice of $h$ and $\beta$.*

*Proof.* The Jacobin of Continuous Sim-HB is written as:

$$\mathcal{J}_S = \begin{bmatrix} \frac{h(1+\beta)}{2(1-\beta)^3} AA^\top & -\frac{1}{1-\beta} A \\ \frac{1}{1-\beta} A^\top & \frac{h(1+\beta)}{2(1-\beta)^3} A^\top A \end{bmatrix}. \tag{47}$$

Let $A = \mathcal{U}\Sigma\mathcal{V}^*$ be the SVD decomposition of the payoff matrix $A$. Thus, $\Sigma$ is a diagonal matrix, and $\mathcal{U}, \mathcal{V}$ are orthogonal matrices. Then the matrix in (47) can be decomposed as

$$\mathcal{J}_S = \begin{bmatrix} \mathcal{U} & \mathbf{0}^\top \\ \mathbf{0} & \mathcal{V}^* \end{bmatrix} \cdot \begin{bmatrix} \frac{h(1+\beta)}{2(1-\beta)^3} \Sigma\Sigma^\top & -\frac{1}{1-\beta} \Sigma \\ \frac{1}{1-\beta} \Sigma^\top & \frac{h(1+\beta)}{2(1-\beta)^3} \Sigma^\top\Sigma \end{bmatrix} \cdot \begin{bmatrix} \mathcal{U}^* & \mathbf{0} \\ \mathbf{0}^\top & \mathcal{V} \end{bmatrix}. \tag{48}$$

Moreover, it is easy to see the matrix $\begin{bmatrix} \mathcal{U} & \mathbf{0}^\top \\ \mathbf{0} & \mathcal{V}^* \end{bmatrix}$ is also an orthogonal matrix. Thus, the eigenvalues of (47) are the same as the eigenvalues of

$$\begin{bmatrix} \frac{h(1+\beta)}{2(1-\beta)^3} \Sigma\Sigma^\top & -\frac{1}{1-\beta} \Sigma \\ \frac{1}{1-\beta} \Sigma^\top & \frac{h(1+\beta)}{2(1-\beta)^3} \Sigma^\top\Sigma \end{bmatrix}. \tag{49}$$

Any eigenvalue $\lambda$ of matrix (49) satisfies the following equation:

$$\left(\lambda - \frac{h(1+\beta)}{2(1-\beta)^3}\rho\right)\left(\lambda - \frac{h(1+\beta)}{2(1-\beta)^3}\rho\right) + \left(\frac{1}{1-\beta}\right)^2 \rho = 0, \tag{50}$$

here $\rho \geq 0$ is an eigenvalue of $\Sigma\Sigma^\top$. The solution of (50) is

$$\lambda = \frac{h(1+\beta)\rho}{2(1-\beta)^3} \pm \sqrt{\Delta}, \tag{51}$$

where $\Delta = -\frac{\rho}{(1-\beta)^2} \leq 0$. Thus, $\sqrt{\Delta}$ is always a purely imaginary number, and

$$\Re(\lambda) = \frac{h(1+\beta)\rho}{2(1-\beta)^3} \geq 0,$$

for any $\beta \in (-1, 1)$ and $h > 0$. Therefore, the equation always diverges at an exponential rate regardless of the choice of $h$ and $\beta$. $\qquad\square$

**Lemma C.2.** *In matrix games $f(x,y) = x^\top A y$, Continuous Alt-HB converges exponentially if $\beta$ is negative and $h$ is sufficiently small.*

*Proof.* The Jacobin of Continuous Alt-HB is written as:

$$\mathcal{J}_A = \begin{bmatrix} \frac{h(1+\beta)}{2(1-\beta)^3} AA^\top & -\frac{1}{1-\beta} A \\ \frac{1}{1-\beta} A^\top & -\frac{h(1-3\beta)}{2(1-\beta)^3} A^\top A \end{bmatrix} \tag{52}$$

Let $A = U\Sigma V$ be the SVD decomposition of the payoff matrix $A$, where $\Sigma$ is a diagonal matrix. Then for a similar reason as in (48), the eigenvalues of matrix in (52) are same as the eigenvalues of

$$
\begin{bmatrix}
\frac{h(1+\beta)}{2(1-\beta)^3}\Sigma\Sigma^\top & -\frac{1}{1-\beta}\Sigma \\[2mm]
\frac{1}{1-\beta}\Sigma^\top & -\frac{h(1-3\beta)}{2(1-\beta)^3}\Sigma^\top\Sigma
\end{bmatrix}.
\tag{53}
$$

Any eigenvalue $\lambda$ of matrix (53) satisfies the following equation:

$$
\left(\lambda - \frac{h(1+\beta)}{2(1-\beta)^3}\rho\right)\left(\lambda + \frac{h(1-3\beta)}{2(1-\beta)^3}\rho\right) + \left(\frac{1}{1-\beta}\right)^2\rho = 0,
\tag{54}
$$

where $\rho \geq 0$ is an eigenvalue of $\Sigma\Sigma^\top$. The solution of (54) are

$$
\lambda = \frac{h\beta\rho}{(1-\beta)^2} \pm \sqrt{\Delta},
\tag{55}
$$

where

$$
\Delta = \frac{h^2\beta^2\rho^2}{(1-\beta)^4} + \frac{h^2(1+\beta)(1-3\beta)\rho^2}{4(1-\beta)^6} - \frac{1}{(1-\beta)^2}\rho.
\tag{56}
$$

From (56), we can choose sufficiently small $h$ so that $\Delta < 0$ because the negative term $-\frac{1}{(1-\beta)^2}\rho$ is independent of $h$. For such small $h$ and negative $\beta$, due to the negativity of $\Delta$, the real part of $\lambda$ is a negative number; therefore, the equation has local convergence. $\square$

## D. Proof of Proposition 4.2 and 4.3

### D.1. Proof of Proposition 4.2

*Proof.* Recall that we define

$$
\mathcal{F}(x, y) = \left(\frac{1}{1-\beta}\right) f(x, y) \\
+ \frac{h(1+\beta)}{4(1-\beta)^3}\left(\|\nabla_x f(x, y)\|^2 - \|\nabla_y f(x, y)\|^2\right),
$$

and the continuous-time model for simultaneous heavy ball method is given by:

$$
\dot{x}(t) = -\nabla_x\mathcal{F}(x, y), \quad \dot{y}(t) = \nabla_y\mathcal{F}(x, y). \tag{Continuous Sim-HB}
$$

The gradient of $\mathcal{F}(x, y)$ with respect to $x$ is calculated as

$$
\nabla_x\mathcal{F}(x, y) = \left(\frac{1}{1-\beta}\right)\nabla_x f(x, y) + \frac{h(1+\beta)}{4(1-\beta)^3}\nabla_x\left(\|\nabla_x f(x, y)\|^2 - \|\nabla_y f(x, y)\|^2\right)
$$

$$
= \left(\frac{1}{1-\beta}\right)\nabla_x f(x, y) + \frac{h(1+\beta)}{2(1-\beta)^3}\left(\nabla_x^2 f(x, y)\cdot\nabla_x f(x, y) - \nabla_{xy}f(x, y)\cdot\nabla_y f(x, y)\right).
$$

The gradient of $\mathcal{F}(x, y)$ with respect to $y$ is calculated as

$$
\nabla_y\mathcal{F}(x, y) = \left(\frac{1}{1-\beta}\right)\nabla_y f(x, y) + \frac{h(1+\beta)}{4(1-\beta)^3}\nabla_y\left(\|\nabla_x f(x, y)\|^2 - \|\nabla_y f(x, y)\|^2\right)
$$

$$
= \left(\frac{1}{1-\beta}\right)\nabla_y f(x, y) + \frac{h(1+\beta)}{2(1-\beta)^3}\left(\nabla_{yx}f(x, y)\cdot\nabla_x f(x, y) - \nabla_y^2 f(x, y)\cdot\nabla_y f(x, y)\right).
$$

The second-order partial derivative matrix $\nabla_x^2 \mathcal{F}(x, y)$ is calculated as

$$
\nabla_x^2 \mathcal{F}(x, y) = \left( \frac{1}{1 - \beta} \right) \nabla_x^2 f(x, y)
$$
$$
+ \frac{h(1 + \beta)}{2(1 - \beta)^3} \left( \nabla_x^3 f(x, y) \cdot \nabla_x f(x, y) + \nabla_x^2 f(x, y) \cdot \nabla_x^2 f(x, y) - \nabla_{xyx} f(x, y) \cdot \nabla_y f(x, y) - \nabla_{xy} f(x, y) \cdot \nabla_{yx} f(x, y) \right).
$$
(57)

Let $(x^*, y^*)$ be the local equilibrium for which we calculate the Jacobian matrix. From the first-order optimality condition of $(x^*, y^*)$, we have the terms $\nabla_x f(x^*, y^*)$ and $\nabla_y f(x^*, y^*)$ in (57) are zero vectors. Thus, we have

$$
\nabla_x^2 \mathcal{F}(x^*, y^*) = \left( \frac{1}{1 - \beta} \right) \nabla_x^2 f(x^*, y^*) + \frac{h(1 + \beta)}{2(1 - \beta)^3} \left( \nabla_x^2 f(x^*, y^*) \cdot \nabla_x^2 f(x^*, y^*) - \nabla_{xy} f(x^*, y^*) \cdot \nabla_{yx} f(x^*, y^*) \right).
$$
(58)

Similarly, we can calculate the term $\nabla_{xy} \mathcal{F}(x^*, y^*)$, $\nabla_{yx} \mathcal{F}(x^*, y^*)$ and $\nabla_y^2 \mathcal{F}(x^*, y^*)$ as follows:

$$
\nabla_{xy} \mathcal{F}(x^*, y^*) = \left( \frac{1}{1 - \beta} \right) \nabla_{xy} f(x^*, y^*) + \frac{h(1 + \beta)}{2(1 - \beta)^3} \left( \nabla_x^2 f(x^*, y^*) \cdot \nabla_{xy} f(x^*, y^*) - \nabla_{xy} f(x^*, y^*) \cdot \nabla_y^2 f(x^*, y^*) \right),
$$
(59)

$$
\nabla_{yx} \mathcal{F}(x^*, y^*) = \left( \frac{1}{1 - \beta} \right) \nabla_{yx} f(x^*, y^*) + \frac{h(1 + \beta)}{2(1 - \beta)^3} \left( \nabla_{yx} f(x^*, y^*) \cdot \nabla_x^2 f(x^*, y^*) - \nabla_y^2 f(x^*, y^*) \cdot \nabla_{yx} f(x^*, y^*) \right),
$$
(60)

$$
\nabla_y^2 \mathcal{F}(x^*, y^*) = \left( \frac{1}{1 - \beta} \right) \nabla_y^2 f(x^*, y^*) + \frac{h(1 + \beta)}{2(1 - \beta)^3} \left( \nabla_{yx} f(x^*, y^*) \cdot \nabla_{xy} f(x^*, y^*) - \nabla_y^2 f(x^*, y^*) \cdot \nabla_y^2 f(x^*, y^*) \right).
$$
(61)

The Jacobian of Continuous Sim-HB at $(x^*, y^*)$ is written as

$$
\mathcal{J}_S = \begin{bmatrix} -\nabla_x^2 \mathcal{F}(x^*, y^*) & -\nabla_{xy} \mathcal{F}(x^*, y^*) \\ \nabla_{yx} \mathcal{F}(x^*, y^*) & \nabla_y^2 \mathcal{F}(x^*, y^*) \end{bmatrix},
$$

where the terms $\nabla_x^2 \mathcal{F}(x^*, y^*), -\nabla_{xy} \mathcal{F}(x^*, y^*), \nabla_y^2 \mathcal{F}(x^*, y^*)$ are given according to (58), (59), (60) and (61).

$$
\mathcal{J}_S = \left( \frac{1}{1 - \beta} \right) \begin{bmatrix} -\nabla_x^2 f(x^*, y^*) & -\nabla_{xy} f(x^*, y^*) \\ \nabla_{yx} f(x^*, y^*) & \nabla_y^2 f(x^*, y^*) \end{bmatrix}
$$
$$
- \frac{h(1 + \beta)}{2(1 - \beta)^3} \begin{bmatrix} -\nabla_x^2 f(x^*, y^*) & -\nabla_{xy} f(x^*, y^*) \\ \nabla_{yx} f(x^*, y^*) & \nabla_y^2 f(x^*, y^*) \end{bmatrix} \cdot \begin{bmatrix} -\nabla_x^2 f(x^*, y^*) & -\nabla_{xy} f(x^*, y^*) \\ \nabla_{yx} f(x^*, y^*) & \nabla_y^2 f(x^*, y^*) \end{bmatrix}
$$
$$
= \left( \frac{1}{1 - \beta} \mathcal{I} - \frac{h(1 + \beta)}{2(1 - \beta)^3} \mathcal{J} \right) \cdot \mathcal{J},
$$

where $\mathcal{I}$ is the identity matrix, and

$$
\mathcal{J} = \begin{bmatrix} -\nabla_x^2 f(x^*, y^*) & -\nabla_{xy} f(x^*, y^*) \\ \nabla_{yx} f(x^*, y^*) & \nabla_y^2 f(x^*, y^*) \end{bmatrix}
$$

is the (Jacobian) for (Gradient Flow). This completes the proof of Proposition 4.2. □

## D.2. Proof of Proposition 4.3

*Proof.* Recall the continuous-time model for alternating heavy ball method is given by:

$$\dot{x}(t) = -\nabla_x \mathcal{F}(x, y),$$
$$\dot{y}(t) = \nabla_y \left( \mathcal{F}(x, y) - \frac{h}{2(1-\beta)^2} \|\nabla_x f(x, y)\|^2 \right). \qquad \text{(Continuous Alt-HB)}$$

With a similar calculation and the use of the first-order optimality condition of local equilibrium $(x^*, y^*)$ presented in Section D.1, the Jacobian of Continuous Alt-HB at $(x^*, y^*)$ can be formulated as

$$\mathcal{J}_A = \mathcal{J}_S - \frac{h}{(1-\beta)^2} \begin{bmatrix} 0 & 0 \\ \nabla_{yx} f \nabla_x^2 f & \nabla_{yx} f \nabla_{xy} f \end{bmatrix}_{(x^*, y^*)}.$$

$\square$

# E. Proof of Theorem 4.6

Recall from the local convergence condition presented in Proposition 2.1, Continuous Sim-HB has local convergence if and only if $\max_{\lambda \in \mathrm{Sp}(\mathcal{J}_S)} \Re(\lambda) < 0$. To analyze $\max_{\lambda \in \mathrm{Sp}(\mathcal{J}_S)} \Re(\lambda)$, we need the following lemma about the eigenvalues of a polynomial matrix.

**Lemma E.1.** *[Graham (2018)] If $p(x)$ is a polynomial and $A \in \mathbb{R}^{n \times n}$, then every eigenvalue of $p(A)$ can be represented by $p(\lambda)$, where $\lambda \in \mathrm{Sp}(\mathcal{J})$.*

*Proof.* The following proof comes from (Graham, 2018). A key point here is that we are considering eigenvalues of matrices over complex number field $\mathbb{C}$, and every $n$-degree polynomial over $\mathbb{C}$ has $n$ roots.

Let $\mu \in \mathbb{C}$ be an eigenvalue of $p(A)$, and we consider the polynomial $p(x) - \mu$. Over the complex number field $\mathbb{C}$, the polynomial $p(x) - \mu$ can be factored as

$$p(x) - \mu = a \prod_{i=1}^{n} (x - \lambda_i) \qquad (62)$$

Moreover, since $\mu$ is an eigenvalue of $p(A)$, we have $\det(p(A) - \mu \mathcal{I}) = 0$. Thus, from (62), as least one of $A - \lambda_i \mathcal{I}$ is not invertible, i.e., $\exists v \in \mathbb{C}^n$, such that $(A - \lambda_i \mathcal{I}) v = 0$. This implies $\mu = p(\lambda_i)$ and finishes the proof. $\square$

In the following, we state a corollary of Assumption 4.5, which is proved in (Wang & Chizat, 2024).

**Lemma E.2.** *[Theorem 2.1 in Wang & Chizat (2024)] Under Assumption 4.5, we have $\Re(\lambda) < 0$ for every $\lambda \in \mathrm{Sp}(\mathcal{J})$.*

Note that the definition of Jacobian for (Gradient Flow) in (Wang & Chizat, 2024) is different from ours by a minus sign. Now we present the proof of Theorem 4.6.

*Proof of Theorem 4.6.* According to Proposition 4.2 and Lemma E.1, every eigenvalue of $\mathcal{J}_S$ can be represented by a quadratic polynomial

$$\left( \frac{1}{1-\beta} - \frac{h(1+\beta)}{2(1-\beta)^3} \lambda \right) \cdot \lambda, \qquad (63)$$

where $\lambda \in \mathrm{Sp}(\mathcal{J})$ is an eigenvalue of (Gradient Flow), and any number represented by (63) is an eigenvalue of $\mathcal{J}_S$.

Thus, to ensure the local convergence of Continuous Sim-HB, we need

$$\Re\left( \left[ \frac{1}{1-\beta} - \frac{h(1+\beta)}{2(1-\beta)^3} \lambda \right] \cdot \lambda \right)$$
$$= \frac{1}{1-\beta} \Re(\lambda) - \frac{h(1+\beta)}{2(1-\beta)^3} \left( \Re(\lambda)^2 - \Im(\lambda)^2 \right) < 0, \quad \forall \lambda \in \mathrm{Sp}(\mathcal{J}). \qquad (64)$$

From Lemma E.2, $\Re(\lambda)$ is a negative number, and from Assumption 4.4, the term $\Re(\lambda)^2 - \Im(\lambda)^2$ in (64) is a negative number. Thus, for some fixed momentum parameter $\beta$ and $\forall \lambda \in \mathrm{Sp}(\mathcal{J})$, we need the step size $h$ in (64) satisfies

$$h < \min_{\lambda \in \mathrm{Sp}(\mathcal{J})} \frac{2(1-\beta)^2}{(1+\beta)} \frac{|\Re(\lambda)|}{(\Im(\lambda)^2 - \Re(\lambda)^2)}, \tag{65}$$

and this finishes the proof of Theorem 4.6. $\qquad\qquad\qquad\qquad\qquad\qquad\qquad\qquad\qquad\square$

## F. Heavy Ball Momentum in Minimization Problems

In this section, we present the interaction between step sizes and momentum parameters of heavy ball momentum methods in minimization. We will see the phenomenon in minimization is significantly different from that in min-max games. The materials in this section are based on (O'Donoghue & Candes, 2015) and (Goh, 2017).

In minimization problems

$$\min_{w \in \mathbb{R}^n} f(w), \tag{Minimization}$$

the heavy ball momentum method is written as

$$w_{t+1} = w_t - \alpha \nabla_w f(w_t) + \beta(w_t - w_{t-1}), \tag{Heavy Ball in Min.}$$

where $\alpha > 0$ is the step size and $\beta \in (0, 1)$ is the momentum parameter.

Let $w^*$ be a local minimum of (Minimization), and without loss of generality, we assume $f(w^*) = 0$. From a local viewpoint, (Minimization) can be approximated by the following quadratic problem near $w^*$:

$$\frac{1}{2} w^\top A w + b^\top w,$$

where $A = \nabla^2 f(w^*)$ and $b = \nabla f(w^*)$. Since $A$ is a symmetric Hessian matrix at a local minimum point, it has an eigenvalue decomposition

$$A = Q \cdot \mathrm{diag}(\lambda_1, .., \lambda_n) \cdot Q^\top,$$

where $\lambda_1 < ... < \lambda_n$.

**Proposition F.1** (O'Donoghue & Candes (2015))**.** *To make* (Heavy Ball in Min.) *achieve local convergence in the minimization problem, we need*

$$0 < \alpha \lambda_n < 2 + 2\beta, \; \beta \in (0, 1),$$

*and therefore a larger momentum in the minimization problem allows the algorithm to achieve local convergence on a boarder range of step sizes.*

*Proof.* The following proof comes from (O'Donoghue & Candes, 2015) and (Goh, 2017).

We introduce an intermediate variable $z_k$ as $z_{k+1} = \beta z_k + \nabla_w f(w_k)$. Then (Heavy Ball in Min.) can be written as

$$z_{k+1} = \beta z_k + \nabla_w f(w_k),$$
$$w_{k+1} = w_k - \alpha z_{k+1}.$$

Let $x_k = Q^\top(w_k - w^*)$ and $y_k = Q^\top z_k$. Then the above update rule can be written as

$$y_{i,k+1} = \beta y_{i,k} + \lambda_i x_{i,k},$$
$$x_{i,k+1} = x_{i,k} - \alpha y_{i,k+1},$$

where $x_{i,k}$ and $y_{i,k}$ represent the $i$-th component of $x_k$ and $y_k$, respectively. These update rules can be written in a matrix iteration form

$$\begin{bmatrix} y_{i,k} \\ x_{i,k} \end{bmatrix} = \mathcal{R}^k \cdot \begin{bmatrix} y_{i,0} \\ x_{i,0} \end{bmatrix},$$

where $\mathcal{R} = \begin{bmatrix} \beta & \lambda_i \\ -\alpha\beta & 1-\alpha\lambda_i \end{bmatrix}$. To make the matrix iteration converge, we need $\max\{|\sigma_1|, |\sigma_2|\} < 1$, where $\sigma_1$ and $\sigma_2$ are the eigenvalues of $\mathcal{R}$. Given this condition and the formulation of $\mathcal{R}$, it turns out that we need

$$0 < \alpha\lambda_n < 2 + 2\beta, \ \beta \in (0,1).$$

$\square$

In the following Figure 7, we present experiments on the dynamical behaviors of heavy ball momentum in minimization. We can observe that small momentum makes the algorithm diverge while large momentum makes the algorithm converge, which supplies Proposition F.1.

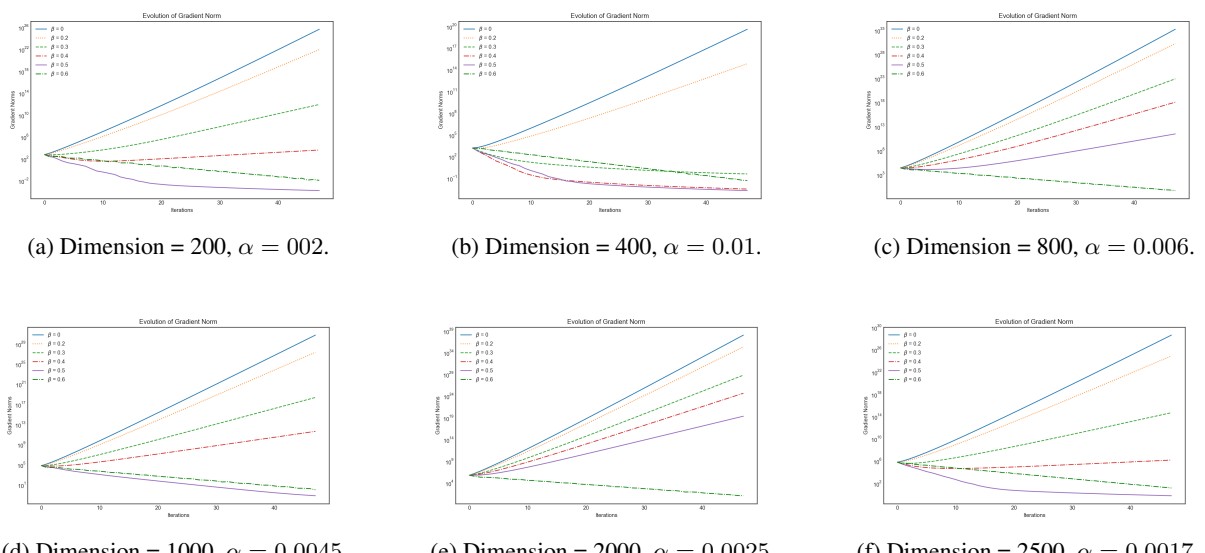

(a) Dimension = 200, $\alpha = 002$.      (b) Dimension = 400, $\alpha = 0.01$.      (c) Dimension = 800, $\alpha = 0.006$.

(d) Dimension = 1000, $\alpha = 0.0045$.      (e) Dimension = 2000, $\alpha = 0.0025$.      (f) Dimension = 2500, $\alpha = 0.0017..$

*Figure 7.* Experimental results for Proposition F.1.

# G. Proof of Theorem 4.8

Recall from Proposition 2.1 and Proposition 4.2, the local convergence rate of Continuous Sim-HB is determined by

$$\max_{\lambda \in \mathrm{Sp}(\mathcal{J})} \frac{1}{1-\beta}\Re(\lambda) + \frac{h(1+\beta)}{2(1-\beta)^3}\left(\Im(\lambda)^2 - \Re(\lambda)^2\right). \tag{66}$$

Moreover, recall that if $f(x,y)$ satisfies Assumption 4.4 and 4.5, then from Lemma E.2, we have $\Re(\lambda) < 0$ and $\Im(\lambda)^2 - \Re(\lambda)^2 > 0$ in (66). Now let $\lambda \in \mathrm{Sp}(\mathcal{J})$, the optimal local convergence rate is achieved by momentum parameter $\beta$ that satisfies the following condition:

$$\mathrm{argmin}_{\beta \in (-1,1)} \left\{ \frac{1}{1-\beta}\Re(\lambda) + \frac{h(1+\beta)}{2(1-\beta)^3}\left(\Im(\lambda)^2 - \Re(\lambda)^2\right) \right\}.$$

In the following, we will explicitly solve this question. Define function

$$g_\lambda(\beta) = \frac{1}{1-\beta}\Re(\lambda) + \frac{h(1+\beta)}{2(1-\beta)^3}\left(\Im(\lambda)^2 - \Re(\lambda)^2\right),$$

and then we can calculate the derivative of $g_\lambda(\beta)$ as:

$$g_\lambda'(\beta) = \frac{\Re(\lambda)}{(1-\beta)^2} + \frac{h\left(\Im(\lambda)^2 - \Re(\lambda)^2\right)(\beta+2)}{(1-\beta)^4}.$$

Thus, we have

$$g'_\lambda(\beta) \geq 0 \Leftrightarrow \Re(\lambda)(1-\beta)^2 + h\left(\Im(\lambda)^2 - \Re(\lambda)^2\right)(\beta+2) \geq 0. \tag{67}$$

Note that the right-hand side of (67) is a quadratic polynomial of $\beta$, which can be written as

$$D_\lambda(\beta) = \Re(\lambda)\beta^2 + \left[h\left(\Im(\lambda)^2 - \Re(\lambda)^2\right) - 2\Re(\lambda)\right]\beta + \Re(\lambda) + 2h\left(\Im(\lambda)^2 - \Re(\lambda)^2\right). \tag{68}$$

When $\beta = 1$, the above function equals to $3h\left(\Im(\lambda)^2 - \Re(\lambda)^2\right)$, which is always a positive number from Assumption 4.4, and when $\beta = -1$, we have

$$D_\lambda(-1) = 4\Re(\lambda) + h\left(\Im(\lambda)^2 - \Re(\lambda)^2\right). \tag{69}$$

Thus, the discussion is decomposed into two cases:

**Case 1, $D_\lambda(-1) > 0$:** In this case, we can conclude that in the whole region $\beta \in (-1, 1)$, $g_\lambda(\beta)$ is an increasing function, and therefore the minimum is taken at point $\beta = -1$. Thus, in this case, the optimal convergence rate is achieved at $\beta = -1$. This case corresponds to the step size $h$ taken in the region

$$h > \frac{4|\Re(\lambda)|}{\Im(\lambda)^2 - \Re(\lambda)^2}$$

and is excluded from the step size region stated as in Theorem 4.8.

**Case 2, $D_\lambda(-1) \leq 0$:** This case covers the step size region

$$h \leq \frac{4|\Re(\lambda)|}{\Im(\lambda)^2 - \Re(\lambda)^2}. \tag{70}$$

For a fixed step size $h$ defined in the region of (70), we denote $\beta(h)$ as the corresponding momentum parameter to achieve the optimal local convergence rate for Continuous Sim-HB. Note that the function $D_\lambda(\beta)$ is a quadratic function of $\beta$, and the coefficient of $\beta^2$-term is $\Re(\lambda)$, which is a negative number according to Lemma E.2. Thus, this optimal $\beta(h)$ should satisfy the condition:

$$D_\lambda(\beta(h)) = 0, \text{ and } \beta(h) \in (-1, 1). \tag{71}$$

From conditions (71), $D_\lambda(1) > 0$ and $D_\lambda(-1) \leq 0$ imply that $\beta(h)$ is the smaller root of $D_\lambda$. Using the roots formula of a quadratic function and $\Re(\lambda) < 0$, we obtain

$$\beta(h) = \frac{2\Re(\lambda) - h\left(\Im(\lambda)^2 - \Re(\lambda)^2\right) + \sqrt{\left[h\left(\Im(\lambda)^2 - \Re(\lambda)^2\right) - 2\Re(\lambda)\right]^2 - 4\Re(\lambda)^2 - 8h\Re(\lambda)\left(\Im(\lambda)^2 - \Re(\lambda)^2\right)}}{2\Re(\lambda)} \tag{72}$$

$$= 1 + \frac{h\left(\Im(\lambda)^2 - \Re(\lambda)^2\right)}{2|\Re(\lambda)|} - \frac{\sqrt{h^2\left(\Im(\lambda)^2 - \Re(\lambda)^2\right)^2 + 12|\Re(\lambda)|h\left(\Im(\lambda)^2 - \Re(\lambda)^2\right)}}{2|\Re(\lambda)|}. \tag{73}$$

From (72), we have

$$\beta(h) > 0 \iff 2 |\Re(\lambda)| + h\left(\Im(\lambda)^2 - \Re(\lambda)^2\right) - \sqrt{h^2 \left(\Im(\lambda)^2 - \Re(\lambda)^2\right)^2 + 12|\Re(\lambda)|h\left(\Im(\lambda)^2 - \Re(\lambda)^2\right)} > 0$$

$$\iff h^2 \left(\Im(\lambda)^2 - \Re(\lambda)^2\right)^2 - 12\Re(\lambda)h\left(\Im(\lambda)^2 - \Re(\lambda)^2\right) < \left(2|\Re(\lambda)| + h\left(\Im(\lambda)^2 - \Re(\lambda)^2\right)\right)^2$$

$$\iff 12|\Re(\lambda)|h\left(\Im(\lambda)^2 - \Re(\lambda)^2\right) < 4|\Re(\lambda)|^2 + 4h|\Re(\lambda)|\left(\Im(\lambda)^2 - \Re(\lambda)^2\right)$$

$$\iff 8h\left(\Im(\lambda)^2 - \Re(\lambda)^2\right) < 4|\Re(\lambda)|$$

$$\iff h < \frac{|\Re(\lambda)|}{2\left(\Im(\lambda)^2 - \Re(\lambda)^2\right)}.$$

Since $\beta(h)$ is the momentum parameter that achieves the optimal local convergence rate for Continuous Sim-HB under step size $h$, this is exactly what we want to prove in Theorem 4.8. Thus, the step size region

$$h < \frac{|\Re(\lambda)|}{2\left(\Im(\lambda)^2 - \Re(\lambda)^2\right)}$$

makes the optimal momentum parameter $\beta(h)$ be a positive number.

# H. Proof of Theorem 4.9

In Section H.1, we provide some notations and lemmas that are useful in the proof of Theorem 4.9. In Section H.2, we provide the proof of Theorem 4.9.

## H.1. Lemmas from first-order matrix perturbation calculation

In this section, we summarize several useful results from matrix perturbation theory, which are important for the proof of Theorem 4.9. Our main reference in this section is (Bamieh, 2020).

Let $\epsilon$ be a small positive number, we consider the behaviors of eigenvalues and eigenvectors of the matrix

$$\mathcal{A}_\epsilon = \mathcal{A}_0 + \epsilon\mathcal{A}_1 \in \mathbb{C}^{n \times n}. \tag{Perturbed Matrix}$$

A vector $v_i \in \mathbb{C}^n$ ($w_i^* \in \mathbb{C}^n$) is a right (left) eigenvector of a matrix $\mathcal{A}$ with eigenvalue $\lambda \in \mathbb{C}$ if $\mathcal{A}v_i = \lambda_i v_i \quad w_i^*\mathcal{A} = \lambda_i w_i^*$. Here we use $w^*$ to represent the conjugate transpose of $w$, i.e., $w^* = \bar{w}^\top$. Given that under the Lebesgue measure on $\mathbb{C}^{n^2}$, almost all matrices over the complex number field $\mathbb{C}$ are diagonalizable, we can use

$$\Lambda = \begin{bmatrix} \lambda_1 & & \\ & \ddots & \\ & & \lambda_n \end{bmatrix} \tag{74}$$

to denote the diagonal matrix consisting of the eigenvalues of $\mathcal{A}$. Moreover, we use

$$\mathcal{V} = [v_1, ..., v_n] \quad \text{and} \quad \mathcal{W} = [w_1, ..., w_n]$$

to denote the matrix consisting of the right and left eigenvectors of $\mathcal{A}$. As a consequence, we have

$$\mathcal{A}\mathcal{V} = \mathcal{V}\Lambda \quad \text{and} \quad \mathcal{W}^*\mathcal{A} = \Lambda\mathcal{W}^*. \tag{75}$$

In (Perturbed Matrix), we assume that the eigenvectors and eigenvalues are analytic functions of $\epsilon$ in a neighborhood of $0$. We use $\mathcal{V}(\epsilon), \mathcal{W}^*(\epsilon)$, and $\Lambda(\epsilon)$ to denote the matrices consisting of the right eigenvectors, left eigenvectors, and eigenvalues

as in (75) and (74). We assume $\mathcal{W}^*(\epsilon), \mathcal{V}(\epsilon)$ have asymptotic expansions as

$$\mathcal{W}^*(\epsilon) = \mathcal{W}_0^* + \epsilon \mathcal{W}_1^* + \dots$$

$$\mathcal{V}(\epsilon) = \mathcal{V}_0 + \epsilon \mathcal{V}_1 + \dots,$$

and then by definition we have

$$(A_0 + \epsilon A_1)(\mathcal{V}_0 + \epsilon \mathcal{V}_1 + \dots) = (\mathcal{V}_0 + \epsilon \mathcal{V}_1 + \dots)(\Lambda_0 + \epsilon \Lambda_1 + \dots) \tag{76}$$

$$(\mathcal{W}_0^* + \epsilon \mathcal{W}_1^* + \dots)(A_0 + \epsilon A_1) = (\Lambda_0 + \epsilon \Lambda_1 + \dots)(\mathcal{W}_0^* + \epsilon \mathcal{W}_1^* + \dots) \tag{77}$$

Moreover, given the potential for an infinite number of choices for $\mathcal{W}^*(\epsilon)$ and $\mathcal{V}(\epsilon)$, it is necessary to impose constraints on these variables. A common choice is the reciprocal basis constraint, which requires that $\mathcal{W}^*(\epsilon)\mathcal{V}(\epsilon) = \mathcal{I}$. This constraint simplifies certain perturbation calculations. As a consequence of $\mathcal{W}^*(\epsilon)\mathcal{V}(\epsilon) = \mathcal{I}$, we need the matrices $\mathcal{V}_0$ and $\mathcal{W}_0^*$ also satisfy the reciprocal basis constraint.

**Lemma H.1** (Bamieh (2020)). *Let $\mathcal{V}_0, \mathcal{W}_0^*$ denote the matrices consisting of the right and left eigenvectors of $\mathcal{A}_0$, and assume they have been normalized according to the reciprocal basis constraint, i.e., $W_0^* V_0 = \mathcal{I}$. Then the diagonal matrix $\Lambda_1$ defined in (76) and (77) can be represented by*

$$\Lambda_1 = \mathrm{dg}\left(\mathcal{W}_0^* \mathcal{A}_1 \mathcal{V}_0\right),$$

*where $\mathrm{dg}(\mathcal{M})$ represents the diagonal elements of matrix $\mathcal{M}$.*

From Lemma H.1, to calculate the first-order eigenvalues, we need to understand the left and right eigenvectors of $\mathcal{A}_0$ satisfy the reciprocal basis constraint.

## H.2. Proof of the theorem

Recall from Proposition 4.3, we have

$$\mathcal{J}_A = \mathcal{J}_S - \frac{h}{(1-\beta)^2} \begin{bmatrix} \mathbf{0} & \mathbf{0} \\ \nabla_{yx} f \nabla_x^2 f & \nabla_{yx} f \nabla_{xy} f \end{bmatrix}_{(x^*, y^*)}. \tag{78}$$

Since $h$ is a small positive number, we will treat the term

$$\frac{h}{(1-\beta)^2} \cdot \begin{bmatrix} \mathbf{0} & \mathbf{0} \\ \nabla_{yx} f \nabla_x^2 f & \nabla_{yx} f \nabla_{xy} f \end{bmatrix}_{(x^*, y^*)}$$

in (78) as a perturbation of the matrix $\mathcal{J}_S$, and use the matrix perturbation theory (Bamieh, 2020) to calculate the eigenvalue of $\mathcal{J}_A$.

Denote

$$\mathcal{P} = \begin{bmatrix} \mathbf{0} & \mathbf{0} \\ \nabla_{yx} f \nabla_x^2 f & \nabla_{yx} f \nabla_{xy} f \end{bmatrix}_{(x^*, y^*)}.$$

This matrix $\mathcal{P}$ will play the role of the matrix $\mathcal{A}_1$ in Section H.1, and the constant $\frac{h}{(1-\beta)^2}$ will play the role of $\epsilon$. According to Lemma H.1, we have

**Lemma H.2.** *Let $\Lambda_A$ denote the diagonal matrix consisting of the eigenvalues of $\mathcal{J}_A$, and $\Lambda_S$ denote the diagonal matrix consisting of the eigenvalues of $\mathcal{J}_S$, then we have*

$$\Lambda_A = \Lambda_S - \frac{h}{(1-\beta)^2} \cdot \mathrm{dg}\left(\mathcal{W}^* \mathcal{P} \mathcal{V}\right) + \mathcal{O}(h^2).$$

*Here $\mathcal{V}, \mathcal{W}^*$ are the matrices consisting of the right and left eigenvector of $\mathcal{J}_S$. Moreover, they need to satisfy the reciprocal basis constraint, i.e., $\mathcal{W}^* \mathcal{V} = \mathcal{I}$.*

From Lemma H.2, the problem of calculating the first-order approximation of the eigenvalues of $\mathcal{J}_A$ is reduced to finding a pair of left and right eigenvectors of $\mathcal{J}_S$ that satisfy the reciprocal basis constraint.

**Lemma H.3.** *Let*

$$\mathcal{J} = \alpha \begin{bmatrix} -\mathcal{Q} & 0 \\ 0^\top & \mathcal{R} \end{bmatrix} + \begin{bmatrix} \boldsymbol{0} & -\nabla_{xy}f \\ \nabla_{yx}f & \boldsymbol{0} \end{bmatrix}_{(x^*,y^*)}, \tag{79}$$

*where $\alpha\mathcal{Q} = \nabla_x^2 f(x^*, y^*)$ and $\alpha\mathcal{R} = \nabla_y^2 f(x^*, y^*)$. Let $\mathcal{V}(\alpha)$ and $\mathcal{W}^*(\alpha)$ be the right and left eigenvectors of $\mathcal{J}$ such that they satisfy the reciprocal basis constraint, i.e., $\mathcal{W}^*(\alpha) \cdot \mathcal{V}(\alpha) = \mathcal{I}$. Then the matrices $\mathcal{W}^*$ and $\mathcal{V}$ in Lemma H.2 satisfy*

$$\mathcal{W}^* = \mathcal{W}^*(\alpha), \quad \mathcal{V} = \mathcal{V}(\alpha).$$

*Proof.* Recall from Proposition 4.2, the Jacobian matrix of Continuous Sim-HB and the Jacobian matrix of (Gradient Flow) satisfies the following quadratic polynomial relation:

$$\mathcal{J}_S = \left( \frac{1}{1-\beta}\mathcal{I} - \frac{h(1+\beta)}{2(1-\beta)^3}\mathcal{J} \right) \cdot \mathcal{J}.$$

Thus, if $v$ is a right eigenvector of $\mathcal{J}$ with respect to an eigenvalue $\lambda$, then we have

$$\mathcal{J}_S v = \left( \frac{1}{1-\beta}\mathcal{I} - \frac{h(1+\beta)}{2(1-\beta)^3}\mathcal{J} \right) \cdot \mathcal{J}v = \left( \frac{1}{1-\beta}\lambda - \frac{h(1+\beta)}{2(1-\beta)^3}\lambda \right) \lambda v.$$

Thus, every right eigenvector of $\mathcal{J}$ is also a right eigenvector of $\mathcal{J}_S$. Similarly, one can prove every left eigenvector of $\mathcal{J}$ is also a left eigenvector of $\mathcal{J}_S$. $\square$

From Lemma H.3 and Lemma H.2, the problem of calculating the first-order approximation of the eigenvalues of $\mathcal{J}_A$ is reduced to finding a pair of left and right eigenvectors of $\mathcal{J}$ in (79) that satisfies the reciprocal basis constraint.

In the following, we will treat the matrix $\mathcal{J}$ in (79) as a perturbation of the matrix

$$\begin{bmatrix} \boldsymbol{0} & -\nabla_{xy}f \\ \nabla_{yx}f & \boldsymbol{0} \end{bmatrix}_{(x^*,y^*)}.$$

If $\mathcal{V}_0$ and $\mathcal{W}_0$ are the matrices consisting of the right and left eigenvectors of $\mathcal{J}$, then from the asymptotic expansion $\mathcal{W}^*(\alpha) = \mathcal{W}_0^* + \mathcal{O}(\alpha)$ and $\mathcal{V}(\alpha) = \mathcal{V}_0 + \mathcal{O}(\alpha)$, we have

$$\mathcal{W}^*(\alpha) \cdot \mathcal{V}(\alpha) = \mathcal{W}_0^* \cdot \mathcal{V}_0 + \mathcal{O}(\alpha) = \mathcal{I},$$

and this requires $\mathcal{W}_0^* \cdot \mathcal{V}_0 = \mathcal{I}$ as $\alpha$ is an arbitrary small number. Here we provide a concrete construction of $\mathcal{V}_0$ and $\mathcal{W}_0$ from the singular value decomposition of $\nabla_{xy}f(x^*, y^*)$.

**Lemma H.4.** *Suppose the matrix $\nabla_{xy}f(x^*, y^*)$ is full-rank and has distinct singular values, and*

$$\nabla_{xy}f(x^*, y^*) = \mathcal{U}\Sigma\mathcal{V}^\top = \sum_{s=1}^{n} \sigma_s u_s v_s^\top$$

*is its singular value decomposition. Then the matrix*

$$\begin{bmatrix} \boldsymbol{0} & -\nabla_{xy}f(x^*, y^*) \\ \nabla_{yx}f(x^*, y^*) & \boldsymbol{0} \end{bmatrix}$$

*has distinct eigenvalues $\{\pm i\sigma_j\}$. Moreover, the right unit-norm eigenvectors of eigenvalue $-i\sigma_j$ is $(-iu_j^\top/\sqrt{2}, v_j^\top/\sqrt{2})^\top$, and the left unit-norm eigenvectors of eigenvalue $i\sigma_j$ is $(iu_j^\top/\sqrt{2}, v_j^\top/\sqrt{2})^\top$.*

*Proof.* This can be verified through a direct calculation.

$$
\begin{bmatrix} \mathbf{0} & -\nabla_{xy}f(x^*,y^*) \\ \nabla_{yx}f(x^*,y^*) & \mathbf{0} \end{bmatrix} \cdot \begin{bmatrix} -\mathrm{i}u_j/\sqrt{2} \\ v_j/\sqrt{2} \end{bmatrix} = \begin{bmatrix} -\nabla_{xy}f(x^*,y^*) \cdot v_j/\sqrt{2} \\ -\nabla_{yx}f(x^*,y^*) \cdot \mathrm{i}u_j/\sqrt{2} \end{bmatrix}
$$

$$
= \begin{bmatrix} -\left(\sum_{s=1}^n \sigma_s u_s v_s^\top\right) \cdot v_j/\sqrt{2} \\ -\left(\sum_{s=1}^n \sigma_s v_s u_s^\top\right) \cdot \mathrm{i}u_j/\sqrt{2} \end{bmatrix} \tag{80}
$$

Moreover, since $\nabla_{xy}f(x^*,y^*) = \mathcal{U}\Sigma\mathcal{V}^\top$ is the SVD of $\nabla_{xy}f(x^*,y^*)$, we have

$$
v_s^\top v_j = \begin{cases} 1, & \text{if } s = j, \\ 0, & \text{if } s \neq j. \end{cases} \quad \text{and} \quad u_s^\top u_j = \begin{cases} 1, & \text{if } s = j, \\ 0, & \text{if } s \neq j. \end{cases}
$$

Thus, (80) can be continued calculated as

$$
\begin{bmatrix} -\left(\sum_{s=1}^n \sigma_s u_s v_s^\top\right) \cdot v_j/\sqrt{2} \\ -\left(\sum_{s=1}^n \sigma_s v_s u_s^\top\right) \cdot \mathrm{i}u_j/\sqrt{2} \end{bmatrix} = \begin{bmatrix} -\sigma_j u_j/\sqrt{2} \\ -\mathrm{i}\sigma_j v_j/\sqrt{2} \end{bmatrix} = -\mathrm{i}\sigma_j \cdot \begin{bmatrix} -\mathrm{i}u_j/\sqrt{2} \\ v_j/\sqrt{2} \end{bmatrix},
$$

this proves $(-\mathrm{i}u_j^\top/\sqrt{2}, v_j^\top/\sqrt{2})^\top$ is an eigenvector of $-\mathrm{i}\sigma_j$. We can prove $(\mathrm{i}u_j^\top/\sqrt{2}, v_j^\top/\sqrt{2})^\top$ is an eigenvector of $\mathrm{i}\sigma_j$ through a similar calculation, and we omit it here. $\square$

**Lemma H.5.** *Denote the matrix*

$$
\mathcal{V}_0 = \left( \begin{bmatrix} -\mathrm{i}u_j/\sqrt{2} \\ \mathrm{i}v_j/\sqrt{2} \end{bmatrix}, \begin{bmatrix} \mathrm{i}u_j/\sqrt{2} \\ \mathrm{i}v_j/\sqrt{2} \end{bmatrix} \right)_{j=1}^n
$$

*and $\mathcal{W}_0^* = \mathcal{V}_0^*$. Then we have $\mathcal{W}_0^* \cdot \mathcal{V}_0 = \mathcal{I}_{2n\times 2n}$, and the matrices $\mathcal{W}(\alpha)$ and $\mathcal{V}(\alpha)$ we are looking for in Lemma H.3 satisfies*

$$
\mathcal{V}(\alpha) = \mathcal{V}_0 + \mathcal{O}(\alpha), \quad \mathcal{W}(\alpha) = \mathcal{W}_0 + \mathcal{O}(\alpha).
$$

*Proof.* Since the matrices $\mathcal{U}$ and $\mathcal{V}$ come from the SVD of $\nabla_{xy}f(x^*,y^*)$, $\mathcal{U}\mathcal{U}^\top$ and $\mathcal{V}\mathcal{V}^\top$ are identity matrices, and this implies $\mathcal{W}_0^* \cdot \mathcal{V}_0 = \mathcal{I}_{2n\times 2n}$. Then this lemma is a direct corollary of Lemma H.3. $\square$

Now we can calculate the eigenvalues of $\mathcal{J}_A$. From Lemma H.2, Lemma H.3 and H.5, we have

$$
\Lambda_A = \Lambda_S - \frac{h}{(1-\beta)^2}\mathrm{dg}\left((\mathcal{W}_0^* + \mathcal{O}(\alpha)) \cdot \mathcal{P} \cdot (\mathcal{V}_0 + \mathcal{O}(\alpha))\right)
$$

$$
= \Lambda_S - \frac{h}{(1-\beta)^2} \begin{bmatrix} w_{0,1}^\top \cdot \mathcal{P} \cdot v_{0,1} & & \\ & \ddots & \\ & & w_{0,2n}^\top \cdot \mathcal{P} \cdot v_{0,2n} \end{bmatrix} + \mathcal{O}(h^2 + \alpha h).
$$

From Lemma H.4, the vectors pair $(\mathcal{W}_{0,j}^\top, \mathcal{V}_{0,j})$ can be chosen as

$$
w_{0,j}^\top = \left(\mathrm{i}u_j^\top/\sqrt{2}, v_j^\top/\sqrt{2}\right) \quad \text{and} \quad v_{0,j}^\top = \left(-\mathrm{i}u_j^\top/\sqrt{2}, v_j^\top/\sqrt{2}\right),
$$

where $u_j$ and $v_j$ come from the singular value decomposition of $\nabla_{xy}f(x^*,y^*)$ as defined in Lemma H.4.

Thus, if $\lambda_A$ is an eigenvalue of $\mathcal{J}_A$, then there exists $\lambda_S \in \mathcal{J}_S$ and a pair of $(u_j, v_j)$, such that

$$\lambda_A = \lambda_S - \frac{h}{(1-\beta)^2} \cdot (\mathrm{i}u_j^\top/\sqrt{2}, \; v_j^\top/\sqrt{2}) \cdot \begin{bmatrix} \mathbf{0} & -\nabla_{xy}f(x^*, y^*) \\ \nabla_{yx}f(x^*, y^*) & \mathbf{0} \end{bmatrix} \cdot \begin{pmatrix} -\mathrm{i}u_j/\sqrt{2} \\ \mathrm{i}v_j/\sqrt{2} \end{pmatrix}$$

$$= \lambda_S - \frac{h}{(1-\beta)^2} \left( v_j^\top \cdot \nabla_{yx}f(x^*, y^*) \cdot \nabla_{xy}f(x^*, y^*) \cdot v_j \right) + \mathcal{O}(h^2 + \alpha h). \tag{81}$$

Moreover, as defined in Lemma H.4,

$$\nabla_{xy}f(x^*, y^*) = \mathcal{U}\Sigma\mathcal{V}^\top = \sum_{s=1}^{n} \sigma_s u_s v_s^\top$$

is the singular value decomposition of $\nabla_{xy}f(x^*, y^*)$. Thus, the term $v_j^\top \cdot \nabla_{yx}f(x^*, y^*) \cdot \nabla_{xy}f(x^*, y^*) \cdot v_j$ in (81) is equal to $|\sigma_j|^2$, and this gives

$$\Re(\lambda_A) = \Re(\lambda_S) - \frac{h}{(1-\beta)^2}|\sigma|^2 + \mathcal{O}(h^2 + h\alpha), \tag{82}$$

which is exactly the formula that we want to prove in Theorem 4.9.

# I. Additional Experiments on Local Convergence Results

If $(x^*, y^*)$ is a local Nash equilibrium of a payoff function $f(x, y)$, then the local structure of $f(x, y)$ is essentially characterized by the following quadratic games:

$$\frac{1}{2}(x - x^*)^\top \nabla_x^2 f(x^*, y^*)(x - x^*) + \frac{1}{2}(y - y^*)^\top \nabla_y^2 f(x^*, y^*)(y - y^*) + (x - x^*)^\top \nabla_{xy} f(x^*, y^*)(y - y^*).$$

(Quadratic Approximation)

Since $(x^*, y^*)$ is a local equilibrium, from its second-order optimality condition, $\nabla_x^2 f(x^*, y^*) \succcurlyeq \mathbf{0}$ is a positive semi-definite matrix and $\nabla_y^2 f(x^*, y^*) \preccurlyeq \mathbf{0}$ is a negative semi-definite matrix.

In the following, we will use (Quadratic Approximation) to test our theoretical results presented in Section 4. In particular, we will test these results using the discrete-time algorithms Sim-HB and Alt-HB. The experimental results show that the theoretical results that we obtain from the continuous-time dynamics can precisely predict the behaviors of these discrete-time algorithms. In the following experiment, we randomly generate $\nabla_x^2 f(x^*, y^*) \succcurlyeq \mathbf{0} \in \mathbb{R}^{n \times n}$, $\nabla_y^2 f(x^*, y^*) \preccurlyeq \mathbf{0} \in \mathbb{R}^{m \times m}$, and $\nabla_{xy} f(x^*, y^*) \in \mathbb{R}^{n \times m}$. Moreover, we allow $\nabla_x^2 f(x^*, y^*)$ and $\nabla_y^2 f(x^*, y^*)$ to have many zero eigenvalues, so that (Quadratic Approximation) is **not** a SCSC function. We achieve this by randomly generating a matrix $A \in \mathbb{R}^{n \times n'}$ with $n' < n$ and setting $\nabla_x^2 f(x^*, y^*) = AA^\top$. Thus, $\nabla_x^2 f(x^*, y^*)$ produced in this way is a positive semi-definite matrix with $n - n'$ zero eigenvalues. Similarly, we can produce $\nabla_y^2 f(x^*, y^*)$ as a negative semi-definite matrix with arbitrary number of zero eigenvalues.

## I.1. Experiments for Theorem 4.6 and Corollary 4.7

The aim of the experiments in this section is to show that smaller momentum can make an algorithm converge across a larger range of step sizes, thus supporting the main results of Theorem 4.6 and Corollary 4.7.

**Experiment 1, 20 dimension problems:** In this experiment, we randomly generate $\nabla_x^2 f(x^*, y^*) \succcurlyeq \mathbf{0} \in \mathbb{R}^{20 \times 20}$, $\nabla_y^2 f(x^*, y^*) \preccurlyeq \mathbf{0} \in \mathbb{R}^{20 \times 20}$, and $\nabla_{xy} f(x^*, y^*) \in \mathbb{R}^{20 \times 20}$. Moreover, we make each of them have 10 zero eigenvalues. The results are presented in the first row of Figure 8. From small to large, the step sizes are chosen as $0.03, 0.08$ and $0.16$. We can observe that smaller momentum can make the algorithm converge under larger step sizes.

**Experiment 2, 200 dimension problems:** In this experiment, we randomly generate $\nabla_x^2 f(x^*, y^*) \succcurlyeq \mathbf{0} \in \mathbb{R}^{20 \times 20}$, $\nabla_y^2 f(x^*, y^*) \preccurlyeq \mathbf{0} \in \mathbb{R}^{20 \times 20}$, and $\nabla_{xy} f(x^*, y^*) \in \mathbb{R}^{200 \times 200}$. Moreover, we make each of them have 100 zero eigenvalues. From small to large, the step sizes are chosen as $0.005, 0.01$ and $0.015$. The results are presented in the second row of Figure 8.

## I.2. Experiments for Theorem 4.8

In this section, we present experiments on Theorem 4.9. Our aim is to show that under small step sizes, positive momentum can achieve a faster convergence rate compared to negative momentum. The results are presented in the first row of Figure 9. Compared with the experiments in I.1, we choose smaller step sizes to guarantee convergence and use larger momentum to compare the convergence rates in the experiments of this section. We can observe that under these small step sizes, there exists a certain algorithm with positive momentum that can achieve the best convergence rate.

**Experiment 1, 20 dimension problems:** In this experiment, we randomly generate $\nabla_x^2 f(x^*, y^*) \succcurlyeq \mathbf{0} \in \mathbb{R}^{20 \times 20}$, $\nabla_y^2 f(x^*, y^*) \preccurlyeq \mathbf{0} \in \mathbb{R}^{20 \times 20}$, and $\nabla_{xy} f(x^*, y^*) \in \mathbb{R}^{20 \times 20}$. Moreover, we make each of them have 10 zero eigenvalues. The results are presented in the first row in Figure 9. From small to large, the step sizes are chosen as $0.003, 0.005$ and $0.008$.

**Experiment 2, 200 dimension problems:** In this experiment, we randomly generate $\nabla_x^2 f(x^*, y^*) \succcurlyeq \mathbf{0} \in \mathbb{R}^{200 \times 200}$, $\nabla_y^2 f(x^*, y^*) \preccurlyeq \mathbf{0} \in \mathbb{R}^{200 \times 200}$, and $\nabla_{xy} f(x^*, y^*) \in \mathbb{R}^{200 \times 200}$. Moreover, we make each of them have 100 zero eigenvalues. The results are presented in the first row of Figure 9. The results are presented in the second row in Figure 9. From small to large, the step sizes are chosen as $0.003, 0.005$ and $0.008$.

## I.3. Experiments for Theorem 4.9

In this section, we provide experiments to verify our theoretical results in Theorem 4.9. Our aim here is to show that for games that satisfy the assumptions in Theorem 4.9, alternating updates can converge faster than simultaneous updates under the same step size and momentum parameter.

**Experiment 1, 100 dimension problems:** In this experiment, we present three randomly generated 100-dimensional games. For each game, we use the same step size $h = 0.01$ and momentum parameter $\beta = 0.4$. The initial conditions are randomly generated. Specifically, we use a small constant number (0.5 here) to scale the matrices $\nabla_x^2 f(x^*, y^*)$ and $\nabla_y^2 f(x^*, y^*)$ to ensure that the settings satisfy the assumptions in Theorem 4.9. In the first row of Figure 10, we present the convergence curves. From these curves, we can observe that alternating updates have a faster convergence rate than simultaneous updates. In the second row, we present the trajectories of the first coordinate variables of the $x$- and $y$-players.

**Experiment 2, 1000 dimension problems:** In this experiment, we present three randomly generated 1000-dimensional games. For each game, we use the same step size $h = 0.001$ and momentum parameter $\beta = 0.2$. The initial conditions are randomly generated. As in previous experiments, we use a small constant number (0.1 here) to scale the diagonal matrices to ensure that the settings satisfy the assumptions in Theorem 4.9. In the first row of Figure 11, we present the convergence curves. From these curves, we can observe that alternating updates have a faster convergence rate than simultaneous updates. In the second row, we present the trajectories of the first coordinate variables of the $x$- and $y$-players. Especially, in the second and third experiments in Figure 11 we can observe a case where alternating updates can converge, while simultaneous updates diverge.

## I.4. Example in Remark 4.10

**Example I.1.** *Let $f(x, y) = ax^2 - by^2 + cxy$, where $a \simeq 2.537$, $b = 0.0003$, and $c = 0.801$. In this example, $a$, $b$, and $c$ correspond to $\alpha Q, \alpha R$, and $P$ in (3). There is a large diagonal element in Jacobian, and therefore the effect of $\alpha$ cannot be ignored. With a choice of $h = 0.4$ and $\beta = 0.2$, the evolution of the distance to equilibrium for simultaneous and alternating updates is presented in Figure 12, where we can observe simultaneous updates converge faster than alternating updates.*

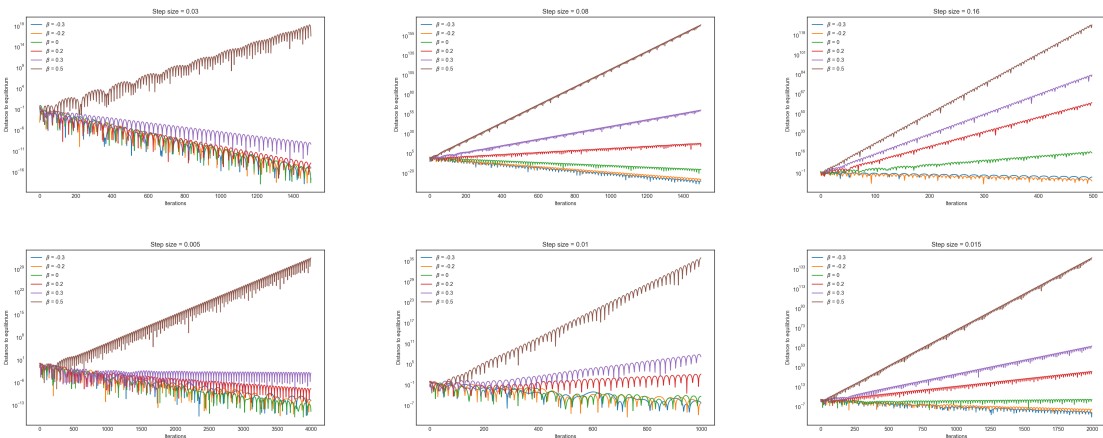

*Figure 8.* Results for experiments on problems with dimension 20 (first row) and dimension 200 (second row). In each row, from left to right, the step sizes increase from small to large.

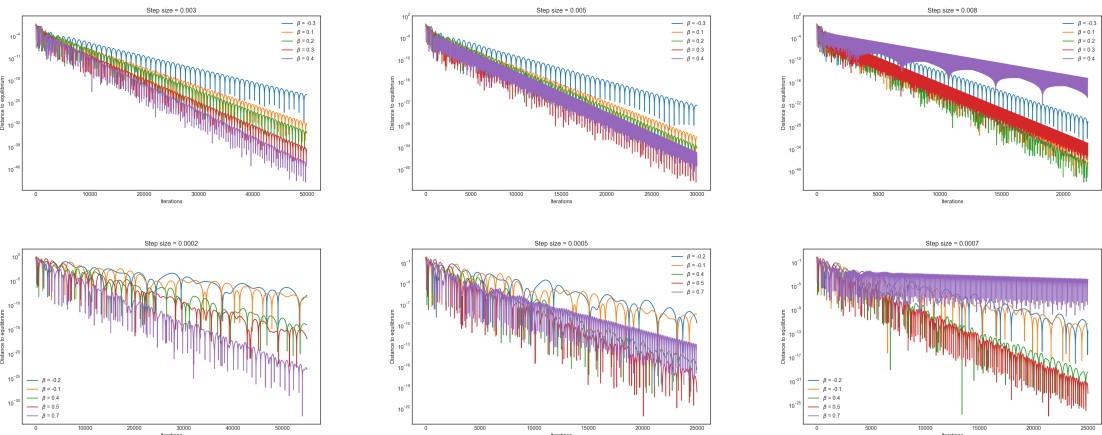

*Figure 9.* Results for experiments on problems with dimension 20 (first row) and dimension 200 (second row). In each row, from left to right, the step sizes increase from small to large.

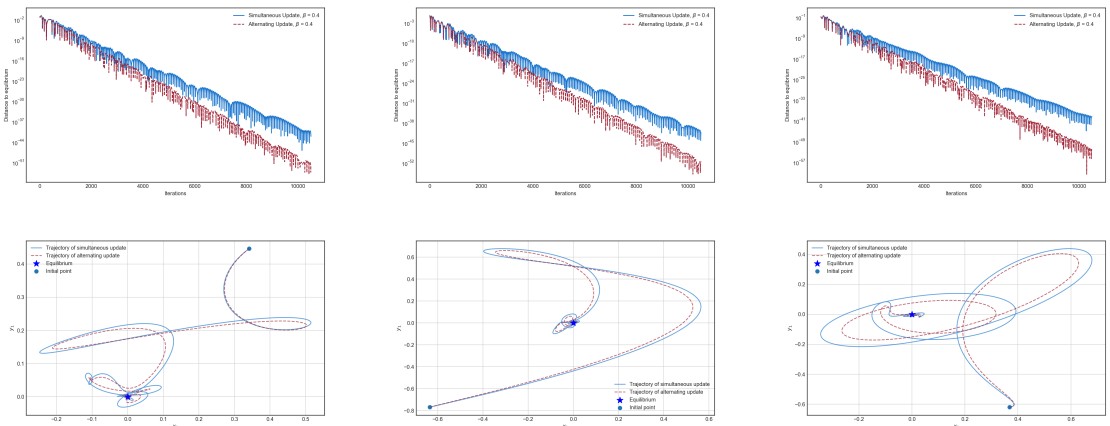

*Figure 10.* Experimental results on problems with dimension 100 for Theorem 4.9.

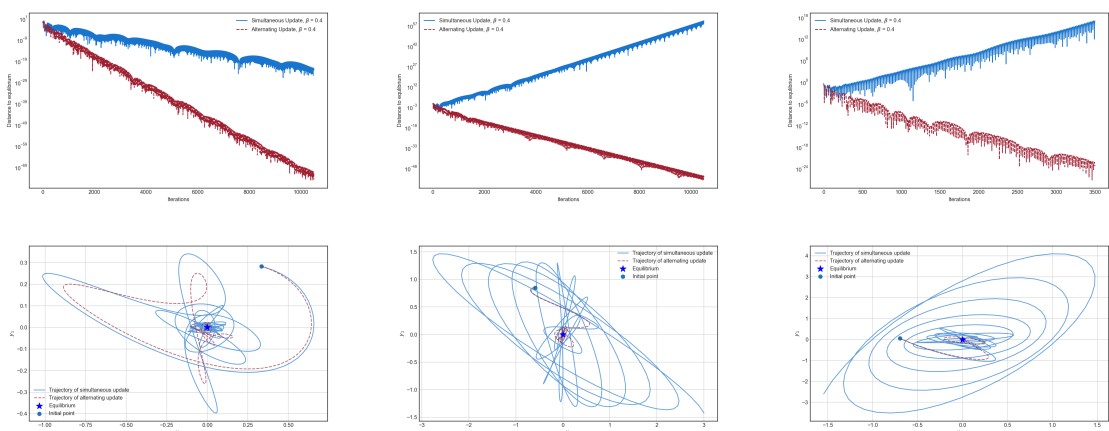

*Figure 11.* Experimental results on problems with dimension 1000 for Theorem 4.9.

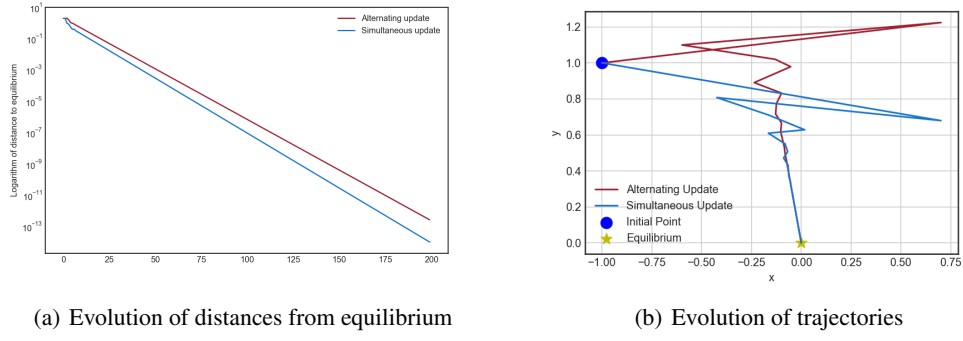

(a) Evolution of distances from equilibrium      (b) Evolution of trajectories

*Figure 12.* Illustration for Example I.1, simultaneous updates converge faster than alternating updates.

# J. Additional Experiments on Implicit Gradient Regularization

### J.1. Adam with negative heavy ball momentum

In Algorithm 1, we provide details of Adam algorithm (Kingma & Ba, 2014) with negative momentum used in the GANs training experiments in Section 5.2. The main difference from the standard Adam algorithm is that the heavy ball momentum parameter $\beta_1$ is chosen as a negative number here.

---

**Algorithm 1** Adam with Negative $\beta_1$

---

1: **Input:**
    Parameters $\theta$; Learning rate $\alpha$;
    $\beta_1 < 0$ **(negative momentum factor; differs from classic Adam)**;
    $\beta_2 \in [0, 1)$; $\varepsilon$ (a small constant)
2: **Output:**
    Updated parameters $\theta$.
3: Initialize $\mathbf{m}_0 \leftarrow \mathbf{0}, \mathbf{v}_0 \leftarrow \mathbf{0}$; $t \leftarrow 0$.
4: **for** each iteration **do**
5:     $t \leftarrow t + 1$
6:     $\mathbf{g}_t \leftarrow \nabla_\theta f(\theta_{t-1})$
7:     $\mathbf{m}_t \leftarrow \beta_1 \mathbf{m}_{t-1} + (1 - \beta_1) \mathbf{g}_t$ {**Key difference:** $\beta_1$ can be negative.}
8:     $\mathbf{v}_t \leftarrow \beta_2 \mathbf{v}_{t-1} + (1 - \beta_2) \mathbf{g}_t^2$
9:     $\hat{\mathbf{m}}_t \leftarrow \frac{\mathbf{m}_t}{1 - \beta_1^t}$
10:     $\hat{\mathbf{v}}_t \leftarrow \frac{\mathbf{v}_t}{1 - \beta_2^t}$
11:     $\theta_t \leftarrow \theta_{t-1} - \alpha \frac{\hat{\mathbf{m}}_t}{\sqrt{\hat{\mathbf{v}}_t} + \varepsilon}$
12: **end for**

---

### J.2. GANs training experiments

**Additional results for Wasserstein GANs.** We firstly present the details of our experimental setting in Section 5.

For Figure 4(b), the experimental setting is:

- Dataset: CIFAR-10.

- Neural network architecture: Both generator and discriminator use the ResNet-32 architecture.

- Learning rate: Both generator and discriminator use the learning rate 2e-4, with a linearly decreasing step size schedule.

- In each training iteration, we update both the discriminator and the generator since in this paper we mainly focus on the setting where both players have the same update steps. Standard Wasserstein GAN training suggests updating the generator every 5 steps (Gulrajani et al., 2017).

- The gradient penalty coefficient is chosen as 10.

In Figure 13, we present the inception scores (Salimans et al., 2016) of the GANs training experiments in the main text. A higher IS indicates that the GANs have better performance. We can observe that lower momentum results in a higher IS. Moreover, compared with simultaneous updates, alternating updates yield higher inception scores. This is consistent with the results reported in the main text and further suggests that the implicit regularization effect is linked to the training quality. Sampled images for alternating training and simultaneous training are presented in Figure 15 and Figure 16. From these pictures, we can observe that the images generated from alternating training look better.

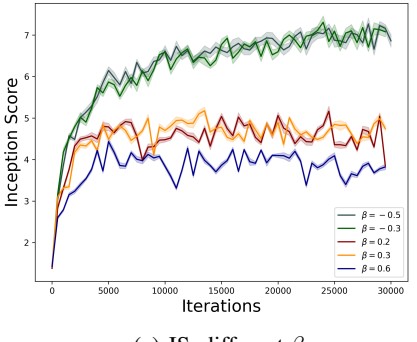

(a) IS, different $\beta$

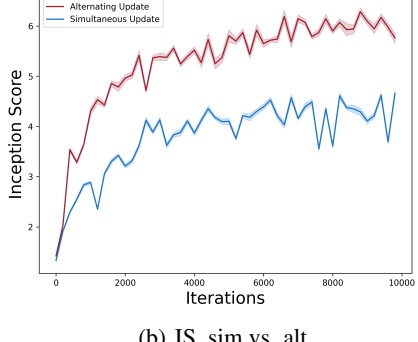

(b) IS, sim vs. alt

*Figure 13.* Inception Scores of Wasserstain GANs training. Smaller momentum leads to a larger inception score, indicating that they have better performance. Moreover, alternating updates lead to a better inception score compared to simultaneous updates.

**Additional results for Vanilla GANs.** In this section we present further results for Vanilla GANs training (Goodfellow et al., 2014). The experimental setting is as follows:

- Dataset: MNIST

- Neural network architecture: MLP

- Learning rate: Both generator and discriminator use the learning rate 0.001,

- In each training iteration, we update both the discriminator and the generator.

- In Figure 14(a), the corresponding heavy ball momentum parameter in Adam is $\beta = 0.6,\ 0.5,\ 0.3,\ 0,\ -0.3,\ -0.5$. In Figure 14(b), both the simultaneous updates and alternating updates use a momentum parameter $\beta = -0.5$.

From Figure 14, we can observe that the experimental results support our thesis regarding the implicit regularization effects in min-max games. Firstly, from Figure 14(a), we can observe that smaller momentum makes the algorithms' trajectories have smaller average slopes. Secondly, from Figure 14(b), we can observe that alternating updates have smaller average slopes compared to simultaneous updates. Since the popular inception score and FID are not suitable for measuring the performance of GANs training on MNIST dataset, we present some sampled images of training with alternating updates and simultaneous updates. In Figure 17, we present sampled images for alternating updates. In Figure 18, we present sampled images for simultaneous updates. We can observe that alternating updates lead to better training results. Especially, in Figure 18 we can observe that under certain parameters, the training has already failed.

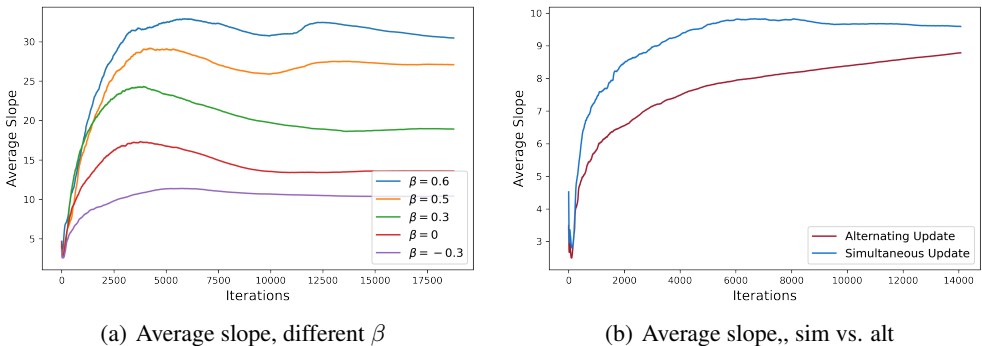

(a) Average slope, different $\beta$        (b) Average slope,, sim vs. alt

*Figure 14.* Average slope of Vanilla GANs training.

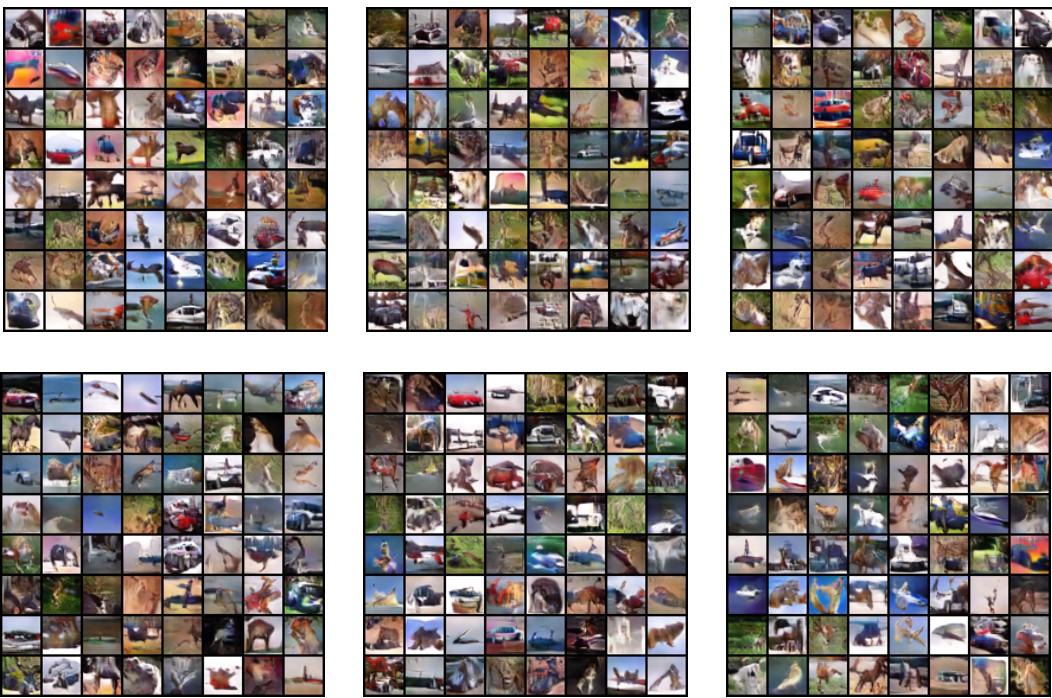

*Figure 15.* Sampled images of Wasserstein GANs with CIFAR-10 dataset, trained by alternating updates.

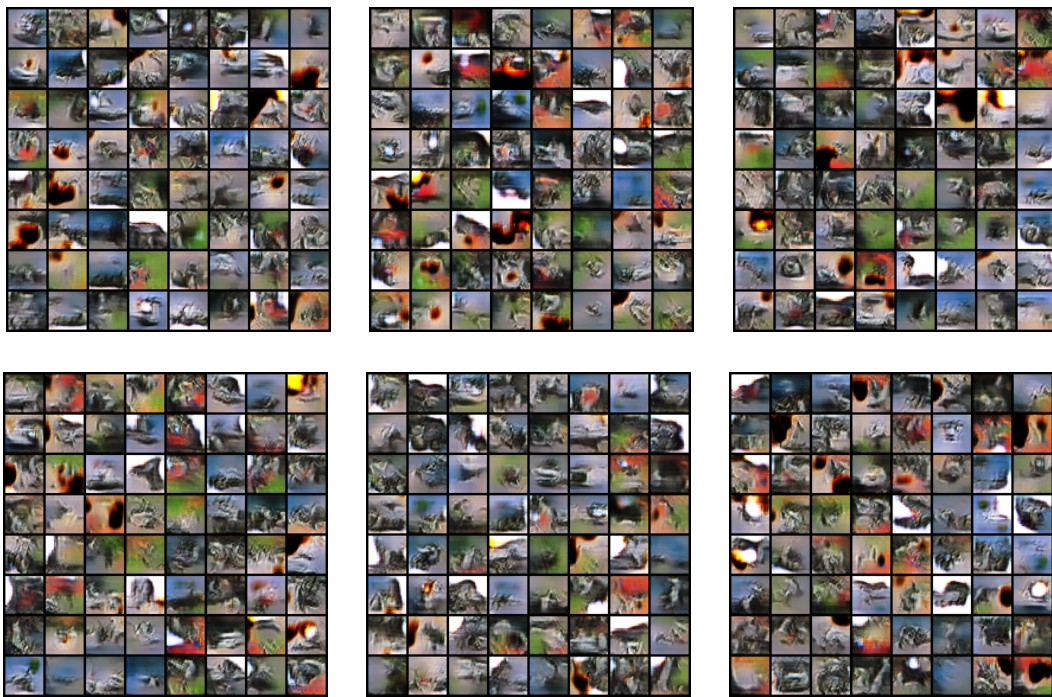

*Figure 16.* Sampled images of Wasserstein GANs with CIFAR-10 dataset, trained by simultaneous updates.

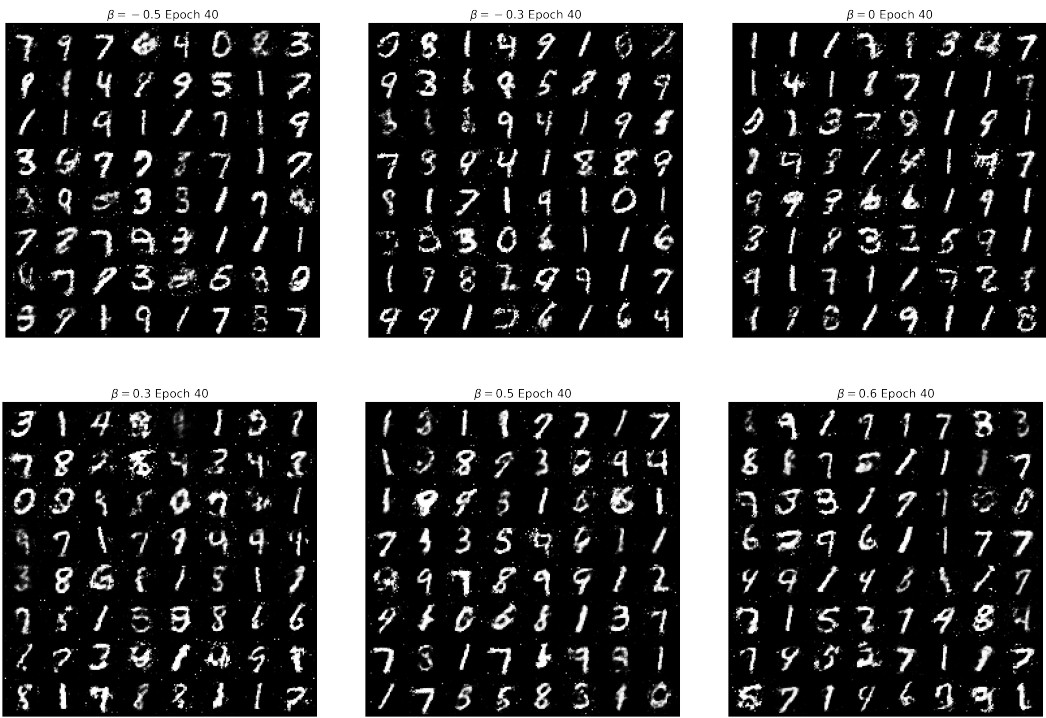

*Figure 17.* Sampled images of Vanilla GANs with MNIST dataset, trained by alternating updates.

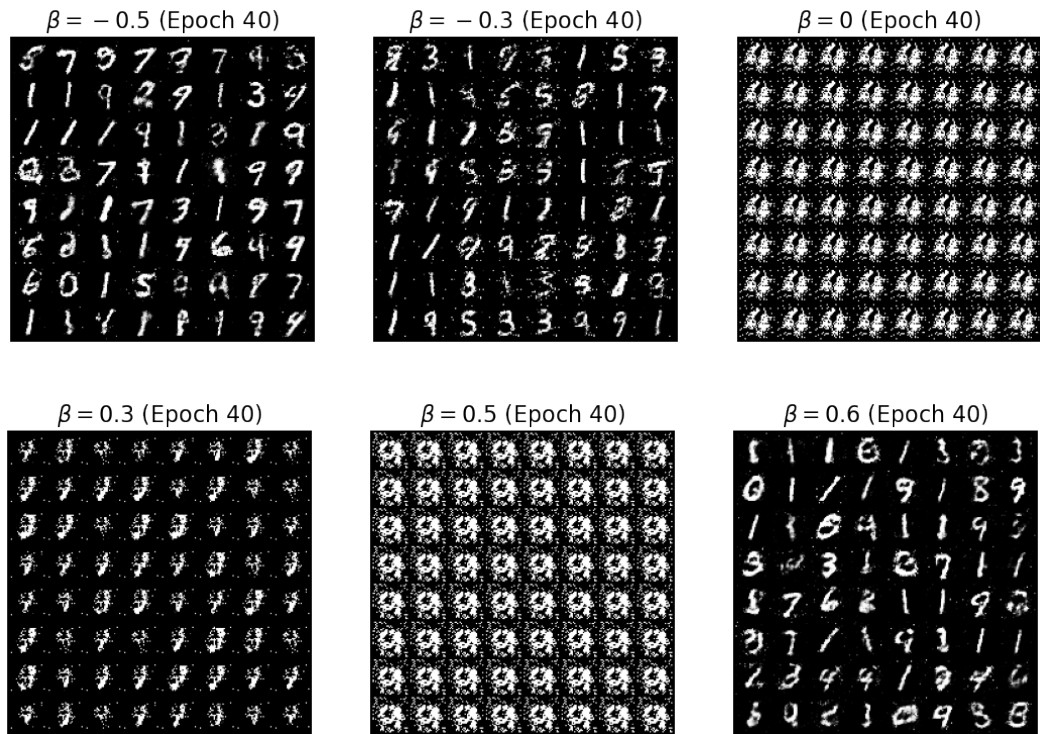

*Figure 18.* Sampled images of Vanilla GANs with MNIST dataset, trained by simultaneous updates.

