# OpenReview forum: "Continuous-Time Analysis of Heavy Ball Momentum in Min-Max Games"
_ICML.cc/2025/Conference — ICML 2025 poster_

### Official Review · Reviewer_HBXK · 2025-02-16

**Overall Recommendation:** 4

**Summary:**

# Summary

This paper explores the role of heavy ball momentum in minmax games via ODEs. While this has been extensively studied in minimization, its effects in minmax is less understood.

## Key Contributions

### 1. Local Convergence Behavior
- **Smaller momentum** improves **stability** and allows for **convergence over a wider range of step sizes**.
- **Alternating updates** generally lead to **faster convergence** than simultaneous updates.
- **Negative momentum** enables convergence even with larger step sizes, a clear difference w.r.t. minimization, where **positive momentum** is typically more successful.

### 2. Implicit Gradient Regularization
- The study reveals that **smaller momentum** encourages optimization trajectories to **prefer shallower slope regions** of the loss landscape.
- Alternating updates **amplify this effect**, stabilizing minmax training.
- This is **opposite w.r.t. what happens in minimization**, where **larger momentum** is associated with improved regularization effects.

### 3. Theoretical and Empirical Validation
- The authors derive ODEs for HB momentum with **simultaneous and alternating update schemes**.
- They establish conditions for **local stability** and **gradient regularization effects**.
- Numerical experiments confirm these findings.

# Update After Rebuttal

Dear Authors,

I am satisfied with your rebuttal, and I am increasing your score to **Accept (4)**: I trust that you will include the enhancements we discussed in the final version of the paper.

Of course, I will follow the discussion with the other Reviewers and AC.

**Claims And Evidence:**

While the theoretical side is sound, the experimental side needs a little enhancement.

When deriving a continuous-time model for an optimizer, I believe it is always crucial to empirically validate that the trajectory of the ODE closely tracks that of the optimizers on a variety of problems. If this is not carefully checked, then one has less guarantee that the theoretical results derived on the ODE actually carry over to the optimizer.

Similarly to [1,2] that derived SDEs to model some optimizers, I suggest that authors plot the dynamics of the two optimizers studied here together with the trajectories of the corresponding ODEs on a two-dimensional game (see Figure 1 in [1] for example).

[1] SDEs for Minimax Optimization, Compagnoni et al., AISTATS 2024.

[2] Sgd in the large: Average-case analysis, asymptotics, and stepsize criticality, Paquette et al., PMLR 2021.

**Essential References Not Discussed:**

As discussed above, it is more about a lack of contextualization in the literature, rather than a lack of a specific paper.
I suggest taking a look at:

1. Weak Approximation Framework: Related Works AND Additional Related Works of [1] for a comprehensive collection of papers that used CTMs to model optimizers.
2. High-Dimensional Setting for SDEs: See [2] and related works.
3. Related Works of [3] focuses on those papers working on CTMs for minimax optimization.

**Experimental Designs Or Analyses:**

The experiments are meaningful and verify their insights. It is unclear whether they used mini batching while training the GANs or not: If this is the case, the results are even stronger. Otherwise, please make it clear AND consider verifying your insights while training with minibatch.

**Methods And Evaluation Criteria:**

N/A

**Other Comments Or Suggestions:**

Based on the weaknesses highlighted above: Studying the stochastic setting is quite crucial in future work because many insights derived in the deterministic setting do not carry over to the stochastic setting. For example, it could be that the negative momentum parameter is detrimental in a stochastic setting, or something along these lines.

This could be fixed by deriving the SDEs of these methods and using Ito calculus to generalize the convergence results.

Regarding the figures: Please, consider enlarging the legend, using different markers for different lines, using a colorblind palette, and so on. While they are quite illustrative, I suggest spending some more time on enhancing them.

# Conclusion

Based on my assessment, this paper is a **Weak Accept**, but I reserve the right to increase to (maximum) **Accept** based on the feedback from the authors and the interaction with the other reviewers and AC.

Necessary conditions are:

1. Enhancing the figures.
2. Expanding the literature review w.r.t. other works using CMTs (ODEs or SDEs) for optimization, with special attention to papers using continuous time models for minimax optimization.
3. Clarifying the experimental setup for the GANs a bit better: I do not want to read the code to figure out the experimental details.
4. Adding some experimental validation that the derived ODEs do track the respective optimizers AT LEAST on popular two-dimensional games.

**Other Strengths And Weaknesses:**

While I really enjoyed reading this well-written paper, I believe that the biggest weakness is the fact that the setup is deterministic and does not handle stochastic gradients. I am quite sure this can be addressed: If not during the rebuttal, maybe in future works. I believe this would be quite relevant because many insights derived in a deterministic setting do not carry over to the stochastic setting.

**Questions For Authors:**

What stopped you from tackling the stochastic setup?

**Relation To Broader Scientific Literature:**

I believe that the authors should dedicate a paragraph of the related works to the use of ODEs and SDEs in optimizations (see Related Works section in [1], which uses the weak approximation framework as well as [2] which tackles this derivation from the high dimensional perspective).

In particular, it is interesting that [3] proved that depending on the game, it might be convenient to select the extrastep of Stochastic Extra Gradient (SEG) to be negative: This somehow reminds me of the negative momentum parameter in this paper. Additionally, [3] also discusses the implicit regularization of SEG and Stochastic Hamiltonian GD: That of SEG is very similar to that of HB, and I believe this should be discussed.

[1] Adaptive Methods through the Lens of SDEs: Theoretical Insights on the Role of Noise, Compagnoni et al., ICLR 2025.

[2] Exact Risk Curves of signSGD in High-Dimensions: Quantifying Preconditioning and Noise-Compression Effects, Xiao et al, arXiv 2025.

[3] SDEs for Minimax Optimization, Compagnoni et al., AISTATS 2024.

**Theoretical Claims:**

I only skimmed the proofs at a high level and they look ok.

---

> ### Author Rebuttal · Authors · 2025-03-31
>
> We sincerely appreciate reviewer's valuable comments and support.  We first respond to the suggestions on the conclusion part of the review:
>
> 1. *Enhancing the figures*
>
> We have updated figures according to your suggestions. Examples are provided through the [anonymous link](https://www.dropbox.com/scl/fi/zmwl1n9l8vpadhf3lc682/Experiments1.pdf?rlkey=8tfl0le2h954qwb0kzy2l95ol&st=cze3w60l&dl=0).
>
> ---
>
> 2. *Expanding the literature review*
>
> We will add an *extended version* of the following paragraph to the **Related Works**.
>
> > **Differential Equations for Optimization.** Differential equations are powerful tools for studying optimization algorithms. In the following, we present a very brief summary, and future related works can be found in the references therein. For minimization, [1,2] purposed ODEs to investigate momentum in convex minimization. The use of SDEs for the stochastic setting was developed by [3] and future developed by [4] recently. For min-max games, [6] introduced methods for deriving high resolution ODEs to study algorithms' convergence behaviors. [5] purposed mean dynamics for Robbins–Monro algorithms to study algorithms' long-time behaviors. For the stochastic setting, recent work of [7] derived SDEs to model the behaviors of several min-max algorithms under the weak approximation framework.
>
> ---
>
> 3. *Experimental setup for the GANs*
>
> We will add the following paragraph into the **Experiments on GANs training** part.
>
> >**Experimental Setup.** The experimental setup generally follows the Wasserstein GANs training framework of (Gulrajani et al., 2017).
> >
> >- Neural network architecture: Both generator and discriminator use the ResNet-32 architecture
> >
> >- Learning rate:  Both generator and discriminator use the learning rate 2e-4, with a linearly decreasing step size schedule
> >
> >- Gradient penalty coefficient: 10
> >
> >- Batch size: 64
> >
> >- During the training, We update both the generator and discriminator in each iteration, which is consistent with the algorithms investigated in this work.
>
> ---
>
> 4. *Experimental validation that the derived ODEs do track the respective optimizers*
>
> We provide experiments in all the three examples provided in Figure 1 of (Compagnoni et al., 2024). Results are provided through the [anonymous link](https://www.dropbox.com/scl/fi/0uhumye9cwlwcd8kftb82/experiments2new.pdf?rlkey=v0c5vfgqgnb984xx6i0o1pogj&st=rhahpsg1&dl=0).
>
> Our continuous-time equations can accurately approximate algorithms' trajectories. For exmaple, for test function 1, trajectories of ODEs and algorithms converge to the same limit cycle. Under the step size $h =0.001$, the maximal errors are around $0.01$ after $10^{5}$ iterations.
>
> ---
>
> ---
>
> In the following, we will address your concerns in the other parts of the review.
>
> 5.  *It is unclear whether they used mini batching while training the GANs or not: If this is the case, the results are even stronger.*
>
> We used the mini batching with batch size of 64 for the experimental results presented in the GANs training (Figure 4). Please refer to Question 3 for future details.
>
> ---
>
> 6.  *... that of SEG is very similar to that of HB ...*
>
> We will add *extended version* of the following discussions in Section 3:
> > It is worth noting that the Hessian-gradient product type of implicit regularization terms in the ODEs for heavy ball momentum is similar to those discovered in the SDEs of Extra-gradient algorithm [7].
>
> ---
>
> 7. *What stopped you from tackling the stochastic setup?*
>
> We specifically study momentum in min-max games under a deterministic setting, following the established line of research that examines momentum in minimization problems under deterministic setting [1,2]. This approach allows us to directly compare the role of momentum in min-max games with its role in minimization, thereby emphasizing their fundamentally different behaviors observed in this paper. We believe that integrating the SDE framework from [7] with our current analysis would be a promising direction for future research. We will highlight this point as a future direction in the conclusion part of this paper.
>
>
> Reference:
>
>
> [1] Su et al., A differential equation for modeling Nesterov's accelerated gradient method: theory and insights, JMLR 2016
>
> [2] Wibisono et al., A variational perspective on accelerated methods in optimization, PNAS 2016
>
> [3] Li et al., Stochastic modified equations and adaptive stochastic gradient algorithms, ICML 2017
>
> [4] Compagnoni et al., Adaptive Methods through the Lens of SDEs: Theoretical Insights on the Role of Noise, ICLR 2025
>
> [5] Hsieh et al., The Limits of Min-Max Optimization Algorithms: Convergence to Spurious Non-Critical Sets, ICML 2021
>
> [6] Lu, An $\mathcal{O}(s^r)$-resolution ODE framework for understanding discrete-time algorithms and applications to the linear convergence of minimax problems, Mathematical Programming (2022)
>
> [7] Compagnoni et al., SDEs for Minimax Optimization, AISTATS 2024

---

> > ### Comment · Reviewer_HBXK · 2025-04-01
> >
> > Dear Authors,
> >
> > I am satisfied with your rebuttal, and I am increasing your score to Accept (**4**): I trust that you will include these enhancements in the final version of the paper.
> >
> > Of course, I will follow the discussion with the other Reviewers and AC.

---

> > > ### Author Response · Authors · 2025-04-02
> > >
> > > Dear Reviewer HBXK,
> > >
> > > It is a great pleasure to hear that you are satisfied with our rebuttal. We will make sure to incorporate the materials from the rebuttal into the revised version of the paper. We sincerely thank you again for your valuable comments.
> > >
> > > Best regards,
> > >
> > > The Authors

---

### Official Review · Reviewer_Fszz · 2025-03-13

**Overall Recommendation:** 4

**Summary:**

This work examines the use of momentum in min-max optimization. The authors investigate both simultaneous gradient descent-ascent (GDA) --as well as it alternating form and their local convergence properties-- plus, heavy ball (HB) momentum. They show that, for simultaneous GDA + HB:
* a positive coefficient achieves optimal convergence,
* while the smaller the momentum coefficient, the broader the range of stable step-sizes.

For alternating GDA + HB:
* they show that there are games with properties around a stationary point such that make alternating GDA + HB converge exponentially faster than GDA + HB.

Further, they empirically demonstrate that smaller negative momentum coefficients lead to stationary points surrounded by areas with generally smaller gradient norms. I.e., stationary points in "flatter" areas of the optimization landscape.

For the purpose of this study, they develop a continuous-time approximation of the heavy-ball method + simultaneous/alternating GDA. This is an approximation that tracks the trajectories of the discrete-time dynamical system by an error that is $O(h^3)$ where $h$ is the step-size.

**Claims And Evidence:**

The claims in the paper are overall founded by rigorous mathematical arguments or empirical evidence. The only claim that could benefit from more extensive experimentation is how flatter saddle-points relate to better generalization. Literature has considered minimization and the loss landscapes of NN optimization but the connection to GANs or other min-max objectives would be interesting.

**Essential References Not Discussed:**

I do not think that they did not discuss some essential reference. Maybe the authors could mention Lu 2022 where they propose a general framework for continuous time ODEs.

Lu, H., 2022. An o (sr)-resolution ode framework for understanding discrete-time algorithms and applications to the linear convergence of minimax problems. Mathematical Programming, 194(1), pp.1061-1112.

**Experimental Designs Or Analyses:**

The authors train a GAN and compare the FID metric to compare different algorithms and hyper-parameters.

**Methods And Evaluation Criteria:**

The methods used make sense for the problem.

**Other Comments Or Suggestions:**

See strengths/weaknesses

**Other Strengths And Weaknesses:**

Strengths:
* The authors carry over an extensive investigation of the local convergence property of the algorithms.
* They contribute a novel ODE for modelling heavy ball momentum and alternating GDA.

Weaknesses:
* The generalization claim is not discussed in an extent, although the better FID scores do demonstrate that the claim has some merit. It is interesting but elaboration is needed.

**Questions For Authors:**

* Do you think that you could get better analysis using the framework proposed in Lu 2022?
* How do you connect the flatness of saddle-points to the flatness of ERM minima in neural nets? Why do you think the GAN gets better FID scores?
* Is there any other implication of the flatness of the minima? How would you connect it to (Ozdaglar et al 2022)?

Lu, H., 2022. An o (sr)-resolution ode framework for understanding discrete-time algorithms and applications to the linear convergence of minimax problems. Mathematical Programming, 194(1), pp.1061-1112.

Ozdaglar, A., Pattathil, S., Zhang, J. and Zhang, K., 2022. What is a good metric to study generalization of minimax learners?. Advances in Neural Information Processing Systems, 35, pp.38190-38203.

**Relation To Broader Scientific Literature:**

The paper relates to optimization theory and specifically min-max optimization. Prior results have have demonstrated scenarios where alternating GDA outperforms simultaneous GDA (Lee 2024).

Also, this paper hints to some similar properties as better generalization when the saddle-points are located in a generally flatter area.

Lee J, Cho H, Yun C. Fundamental benefit of alternating updates in minimax optimization.
Wang, J.-K., Lin, C.-H., Wibisono, A., and Hu, B. Provable acceleration of heavy ball beyond quadratics for a class of Polyak-Lojasiewicz functions when the non-convexity is averaged-out.
Wibisono, A., Tao, M., and Piliouras, G. Alternating mirror descent for constrained min-max games.
Hochreiter, S. and Schmidhuber, J., 1997. Flat minima.

**Theoretical Claims:**

I checked the theoretical claims and their proofs.

---

> ### Author Rebuttal · Authors · 2025-03-31
>
> We sincerely appreciate reviewer's valuable comments and support. Please see our itemized responses below:
>
> 1. *Do you think that you could get better analysis using the framework proposed in Lu 2022?*
>
> We thank the reviewer for highlighting the potential connection between our current paper and (Lu 2022). We will add (Lu 2022) to the **Related Works** section. We believe that exploring the possibility of combining the $\mathcal{O}(r^s)$-resolution technique of (Lu 2022) with heavy ball momentum is a highly promising direction. This could potentially lead to continuous-time equations for heavy ball momentum in min-max games with improved accuracy. In particular, the global convergence property of the ODE proposed by (Lu 2022) would be especially interesting if it can be applied to study the heavy ball momentum algorithm, whose global convergence behavior in general min-max games remains unclear.
>
> ---
>
> 2. *How do you connect the flatness of saddle-points to the flatness of ERM minima in neural nets? Why do you think the GAN gets better FID scores?*
>
> We thanks the reviewer for raising this important question. As the experiments in the current paper suggest, the superior performance of GANs trained with negative momentum is related to their implicit regularization effect. This effect guides the algorithm's trajectory to explore regions with shallower slopes in the GANs loss landscape. However, we would like to emphasize that training dynamics of min-max games exhibit subtle differences compared to ERM minima in neural networks. One notable distinction is that first-order methods are guaranteed to converge to local minima in ERM problems [1]. In contrast, for min-max games, such methods might lead the algorithms to converge to limited sets, such as cycles, rather than saddle points [2]. Therefore, we believe the relationship between the "flatness" of the min-max loss landscape and its machine learning applications could be more complex and multifaceted than in minimization problems. We also highlight building a solid theoretical understanding of this relationship as an important future research direction in the conclusion section of the current paper.
>
> From a high-level perspective, we believe that the shallower slope regions of the GANs loss landscape may represent a certain level of "**robustness**". In these regions, perturbations to the parameters of the generator and discriminator are less likely to significantly impact the values of their loss functions. This offers a potential explanation for why parameters from the shallower slope regions of the GANs loss landscape tend to perform better. We believe that further exploration in this area could be an interesting direction.
>
> ---
>
> 3. *Is there any other implication of the flatness of the minima? How would you connect it to (Ozdaglar et al 2022)?*
>
> From the experiments presented in the current paper, we observe that shallower slope regions in the GANs loss landscape tend to exhibit better performance when measured by FID and Inception Score. These two metrics evaluate the quality, diversity, and similarity of individual samples between generated and real data. Therefore, we believe that in the context of GANs, flatness can bring benefits in these aspects. We also believe that further exploring the implications of the flatness of the min-max loss landscape in other applications could be an intriguing area for future research.
>
> We thank the reviewer for pointing out the literature by (Ozdaglar et al 2022). We find the metric of primal gap to study generalization of minimax algorithms purposed by (Ozdaglar et al 2022) is particularly insightful. It would be interesting to explore whether these tools could be connected to the flatness property of the loss landscapes. Additionally, we notice that the theoretical results in (Ozdaglar et al. 2022) are primarily proven for non-convex-concave or convex-concave cases, which differ from the non-convex-non-concave nature of GANs or other pratical applications of min-max games in machine learning, like adversarial training. We believe there is significant potential for further exploration in this area.
>
> Reference:
>
> [1] Lee et al., First-order Methods Almost Always Avoid Saddle Points, Mathematical Programming 2019
>
> [2] Hsieh et al., The Limits of Min-Max Optimization Algorithms: Convergence to Spurious Non-Critical Sets, ICML 2021

---

> > ### Comment · Reviewer_Fszz · 2025-04-04
> >
> > I thank the authors for their extensive reply. I would encourage them to include an extended discussion of the relationship between the flatness of saddle-points and generalization and connections to robust ML. I think it would help with the paper's dissemination and contribute to the topic of the implicit bias and generalization properties of models trained using min-max optimization. Also, you could discuss meta-learning in games properties [1] of momentum based algorithms.
> >
> >
> > Good luck
> >
> > ---
> > [1] Harris, K., Anagnostides, I., Farina, G., Khodak, M., Wu, Z.S. and Sandholm, T., 2022. Meta-learning in games.

---

> > > ### Author Response · Authors · 2025-04-05
> > >
> > > Dear Reviewer Fszz,
> > >
> > > We sincerely thank you once again for your valuable suggestions. We will include an extended discussion on the flatness of saddle points and its potential implications in the revised version of the paper. We will also discuss several related papers as suggested in your review.
> > >
> > > Best regards,
> > >
> > > The Authors

---

### Official Review · Reviewer_thnv · 2025-03-19

**Overall Recommendation:** 2

**Summary:**

This paper investigates the role of Heavy Ball (HB) momentum in min-max games, an area that has been largely unexplored compared to its well-studied application in minimization. In order to analyze Heavy Ball method, the authors follow a continuous dynamics approximation of the algorithm for simultaneous & alternative version.

For **Sim-HB**, the continuous-time models are given by:

$
\dot{x}(t) = -\nabla_x \mathcal{F}(x,y), \quad \dot{y}(t) = \nabla_y \mathcal{F}(x,y).
$

*(Continuous Sim-HB)*

---

For **Alt-HB**, the continuous-time models are:

$
\dot{y}(t) = \nabla_y \left( \mathcal{F}(x,y) - \frac{h}{2(1 - \beta)^2} \|\nabla_x f(x,y)\|^2 \right),
$

$
\dot{x}(t) = -\nabla_x \mathcal{F}(x,y).
$

*(Continuous Alt-HB)*

**Claims And Evidence:**

•	Local Analysis: The study finds that smaller momentum improves algorithmic stability, allowing for convergence across a broader range of step sizes. Alternating updates lead to faster convergence than simultaneous updates.

•	Global Analysis: Smaller momentum implicitly regularizes the optimization process, guiding algorithm trajectories toward shallower slope regions of the loss landscape. Alternating updates further amplify this effect.

•	Key Insight: These findings contrast with standard minimization, where larger momentum typically improves convergence. This reveals fundamental differences in how HB behaves in min-max games versus standard minimization problems.


One of the primary concerns here is discerning the true value of these results and their broader significance. While this may be discussed in further detail elsewhere, it is imperative to assess what has genuinely been gained.

The first notable achievement is the avoidance of the classical second-order differential equation. The authors assert that this enhances the approximation of the error rate. However, the more critical issue is the lack of a lemma demonstrating that this analysis yields a clearer proof for the discrete algorithm. This omission is particularly striking, as there is already a remark suggesting that the existing work of Gidel provides an analysis for the discrete case. Consequently, what we have here is merely a continuous dynamical system that aligns with the observed behaviour, rather than offering an avenue to deduce insights about the discrete counterpart. The reverse approach—deriving from the continuous system a clean and direct understanding of the discrete case—would have been of far greater interest.

**Essential References Not Discussed:**

N/A

**Experimental Designs Or Analyses:**

Exposition of theoretical claims

**Methods And Evaluation Criteria:**

N/A

**Other Comments Or Suggestions:**

The crucial issue here is to clarify which aspects of the continuous analysis can be transferred in a black-box manner to the discrete case. Without this, the practical implications of the theoretical framework remain uncertain.

**Other Strengths And Weaknesses:**

I will leave it to the area chair to determine whether this constitutes a strength or a weakness.

The paper certainly aims to discuss the benefits of the heavy-ball method through the lens of continuous dynamics, employing a modified first-order method to achieve an $h^3$ approximation. However, the most significant shortcoming is that, particularly in the implicit regularization section, there is no clear mathematical statement supporting the claims and did not show the proof-connection between discrete and continuous case.

**Questions For Authors:**

Please respond to my concerns in the multiple sections of my review

**Relation To Broader Scientific Literature:**

Let me begin with a minor issue: I was unable to find in Bailey 2020 the claim stating that Sim-GDA cycles and does not diverge. In fact, I believe the paper explicitly asserts the opposite—that Sim-GDA diverges, and even Alt-GDA exhibits similar behavior.

However, the more pressing concern is that momentum with a different parameter setting is, in essence, a variation of Optimistic Gradient Descent (OGD). Yet, I did not see a clear discussion on this aspect, nor any broader consideration of learning-augmented algorithms that incorporate predictions within the gradient oracle framework. This omission is particularly notable, as it leaves a key conceptual connection unexplored.

**Theoretical Claims:**

Yes

---

> ### Author Rebuttal · Authors · 2025-03-31
>
> We sincerely appreciate the reviewer's valuable comments. Regarding the concern about the practical implications of the results from continuous-time equations (CTEs), we provide a clearer explanation of the importance and relevance of continuous-time analysis:
>
> *Continuous-time analysis is a **widely accepted methodology** for analyzing optimization algorithms. It not only provides **novel insights** into the behaviors of corresponding algorithms but also proves to be **indispensable** in certain contexts.*
>
> We now turn to a detailed discussion.
>
> >**Widely Accepted Methodology**: We list several papers that have a similar approach to our paper: rather than analyzing the original algorithms, they derive well-designed CTEs for the algorithms and analyze these equations to understand the algorithms' behavior. In minimization, these papers include [1,2,3]. In min-max games setting, papers include [4,5,6]. *Although the results in these papers only obtained rigorous proof for CTEs, they are also crucial for understanding the behavior of the original algorithms and are well accepted by the community.*
>
> >**Novel Insights**: As the reviewer noted, the results in this paper *"reveals fundamental differences in how HB behaves in min-max games versus standard minimization problems."* We sincerely appreciate this observation and would like to emphasize that it is precisely the insights gained through analyzing the CTEs that lead us to discover such novel phenomena. While the reviewer comments that *" what we have here is merely a continuous dynamical system that aligns with the observed behaviour"*, we respectfully suggest that this alignment underscores the strength of well designed CTEs in uncovering novel insights into original algorithms.
>
> >**Indispensable**: CTEs are indispensable for studying implicit gradient regularization effects (IGRs), which is the focus of Section 5 in our paper. The foundational work in this area [7] introduced CTEs for gradient descent, demonstrating that these algorithms implicitly favor flat minima, which are referred to as IGRs. *IGRs only become apparent through the analysis of the regularization terms present in the CTEs*. Several subsequent works, including [4,8], have also relied on the same approach.
>
> We fully agree with the reviewer's point that deriving results for the original algorithms is important, and we will list it as a future work. At the same time, we hope the above discussion demonstrates that results obtained from continuous-time analysis play a crucial role in understanding algorithmic behavior.
>
> Below, we address additional concerns raised by the reviewer.
>
> 1. *" ...  as there is already a remark suggesting that the existing work of Gidel provides an analysis for the discrete case..."*
>
> Due to the nature of the proof technique they adopted, the results of Gidel et al. for discrete-time algorithms are **only** applicable to bilinear games, which is a special case of the games considered in our paper.
>
> ---
>
> 2. *"... momentum with a different parameter setting is, in essence, a variation of Optimistic Gradient Descent (OGD) ..."*
>
> We thank the reviewer for pointing out the omission of the discussion in OGD. OGD and HB methods are based on different approaches. OGD is to *"incorporate predictions within the gradient oracle framework"*. In contrast, the mechanism of momentum methods focuses on simulating specific physical processes [9]. These differences are also reflected by the qualitative difference of the two approaches in simple bilinear games, Sim-HB diverge while Sim-OGD converge.
>
> In the refined version we will incorporate a detailed discussion on the differences between OGD and HB.
>
> ---
>
> 3. *"...implicit regularization section, there is no clear mathematical statement ..."*
>
> Due to the complexity of the optimization dynamics, findings in the research area of implicit gradient regularization are presented as **qualitative descriptions**. For example, in previous work [7] of this direction, the results are formulated as a *Prediction* supported by experimental results. Similarly, we state our results as a *Thesis*, and support with by experimental results.
>
> Reference:
>
> [1] Kovachki & Stuart, Continuous-Time Analysis of Momentum Methods, JMLR 2020
>
> [2] Muehlebach & Jordan, Continuous-Time Lower Bounds for Gradient-based Algorithms, ICML 2020
>
> [3] Romero & Benosman, Finite-Time Convergence in Continuous-Time Optimization, ICML 2020
>
> [4] Rosca et al., Discretization Drift in Two-Player Games, ICML 2021
>
> [5] Hsieh et al., The Limits of Min-Max Optimization Algorithms: Convergence to Spurious Non-Critical Sets, ICML 2021
>
> [6] Compagnoni et al, SDEs for Minimax Optimization, AISTATS 2024
>
> [7] Barrett & Dherin, Implicit Gradient Regularization, ICLR 2021
>
> [8] Ghosh et al., Implicit Regularization in Heavy-ball Momentum Accelerated Stochastic Gradient Descent, ICLR 2023
>
> [9] Qian, On the Momentum Term in Gradient Descent Learning Algorithms, Neural Networks 1999

---

### Decision · Program_Chairs · 2025-05-01

**Decision:**

Accept (poster)

**Comment:**

Summary: The paper studies a continuous-time analysis for HB with simultaneous and alternating update schemes in min-max games. Locally, the paper proves that a smaller momentum improves algorithmic stability by enabling local convergence across a wider range of step sizes, with alternating updates generally converging faster. Globally, a smaller momentum guides algorithms' trajectories towards shallower slope regions of the loss landscapes, with alternating updates amplifying this effect. The proposed results of the paper reveal fundamental differences between HB in min-max games and minimization, and numerical experiments further validate our theoretical results.

On reviews: The reviewers agreed that the paper is well written and has clear contributions. They highlight the extensive investigation of the local convergence property of the algorithms and the novel ODE for modelling heavy ball momentum and alternating GDA.

Recommendation:
I strongly advise the authors to make all agreed-upon changes in the updated version of their work, in terms of presentation and experiments. In particular,

- Enhancing the figures.
- Expanding the literature review w.r.t. other works using CMTs (ODEs or SDEs) for optimization, with special attention to papers using continuous time models for minimax optimization.
- Clarifying the experimental setup for the GANs a bit better: I do not want to read the code to figure out the experimental details.
- Adding some experimental validation that the derived ODEs do track the respective optimizers AT LEAST on popular two-dimensional games.

It would also be interesting to explore the stochastic regime in future works.

I suggest acceptance, as I trust the authors will make the necessary updates.